# Small object detection algorithm incorporating swin transformer for tea buds

**Meiling Shi**[1], **Dongling Zheng**[1], **Tianhao Wu**[2], **Wenjing Zhang**[1], **Ruijie Fu**[1], **Kailiang Huang**[2]*

**1** College of Information Engineering, Sichuan Agricultural University, Ya'an, China, **2** College of Electrical Engineering, Sichuan Agricultural University, Ya'an, China

* hkailiang@163.com

**Data Availability Statement:** All data files are available from the kaggle database (URLs https://www.kaggle.com/datasets/meiling12/tea-budsyolo).

## Abstract

Accurate identification of small tea buds is a key technology for tea harvesting robots, which directly affects tea quality and yield. However, due to the complexity of the tea plantation environment and the diversity of tea buds, accurate identification remains an enormous challenge. Current methods based on traditional image processing and machine learning fail to effectively extract subtle features and morphology of small tea buds, resulting in low accuracy and robustness. To achieve accurate identification, this paper proposes a small object detection algorithm called STF-YOLO (Small Target Detection with Swin Transformer and Focused YOLO), which integrates the Swin Transformer module and the YOLOv8 network to improve the detection ability of small objects. The Swin Transformer module extracts visual features based on a self-attention mechanism, which captures global and local context information of small objects to enhance feature representation. The YOLOv8 network is an object detector based on deep convolutional neural networks, offering high speed and precision. Based on the YOLOv8 network, modules including Focus and Depthwise Convolution are introduced to reduce computation and parameters, increase receptive field and feature channels, and improve feature fusion and transmission. Additionally, the Wise Intersection over Union loss is utilized to optimize the network. Experiments conducted on a self-created dataset of tea buds demonstrate that the STF-YOLO model achieves outstanding results, with an accuracy of 91.5% and a mean Average Precision of 89.4%. These results are significantly better than other detectors. Results show that, compared to mainstream algorithms (YOLOv8, YOLOv7, YOLOv5, and YOLOx), the model improves accuracy and F1 score by 5-20.22 percentage points and 0.03-0.13, respectively, proving its effectiveness in enhancing small object detection performance. This research provides technical means for the accurate identification of small tea buds in complex environments and offers insights into small object detection. Future research can further optimize model structures and parameters for more scenarios and tasks, as well as explore data augmentation and model fusion methods to improve generalization ability and robustness.

**Funding:** This work was supported by the Subsidy for University Student Entrepreneurship Training Program (No. 202310626003).

**Competing interests:** The authors have declared that no competing interests exist.

## Introduction

Tea, as one of the world's three principal beverages, is universally esteemed and sought after by nations across the globe. In recent years, individuals' interest in tea has transcended beyond its mere flavor, delving into its nutritional and medicinal virtues. With over 50 countries, including China, India, and Vietnam, engaged in tea production on a global scale, the industry has significantly bolstered the economies of several tea-cultivating nations in Asia and Africa [1]. In 2020, global tea production reached an impressive 6,269,000 tonnes, with the worldwide tea cultivation area expanding to 5,098,000 hectares. Despite these strides, the tea industry's growth has been curtailed by the challenges of labor recruitment and the escalating costs of labor [2]. Labor dedicated to the picking of tea buds constitutes 60% of the workforce employed in the comprehensive management of tea plantations. To address this labor-intensive issue, artificial intelligence algorithms have been synergized with machinery to facilitate intelligent picking. However, the diverse positions, postures, and densities at which tea buds grow pose a significant challenge to mechanized picking, particularly in complex environments characterized by wind and fluctuating light conditions [3]. In recent years, with the advancement of computer vision technology, numerous network models boasting high precision and real-time advantages have emerged. These high-performance models have been widely applied across various fields, achieving remarkable results and providing technical support for the realization of intelligent tea picking. Therefore, an effective approach to ensuring the excellence of the tea production line is to accurately identify and pick tea buds.

So far, there have been many researchers who have contributed to the field of tea object detection. Xu, Wenkai, et al. [3] proposed an approach for detecting and classifying tea buds that combines the fast YOLOv3 network with the highly accurate DenseNet 201. One of the biggest highlights of the study is the use of datasets taken from two different viewpoints: the side view and the top view. Interestingly, the accuracy of the datasets obtained from the side view is higher than that of the top view on the proposed network. This finding provides valuable empirical guidance for the subsequent production of the tea bud dataset. However, it is important to note that the study overlooks the potential impact of the shooting distance and the size of the detected object on the accuracy of the results. Xie, Shuang, and Hongwei Sun [3] proposed the Tea-YOLOv8s network to reduce interference from complex backgrounds in detecting tea buds, resulting in improved precision. However, the mean average precision (mAP) was not sufficiently high, reaching only 88.27%. Xue, Zhenyang, et al. [4] proposed a method that utilizes YOLO-based networks and incorporates modules such as attention mechanisms. However, their approach still faced limitations in achieving high precision and effectively addressing challenges such as leaf occlusion, lighting conditions, and the detection of extremely small objects. Wu, Yanxu, et al. [5] pointed out that existing models are based on RGB images, limiting the detection of partial information. They utilised the multimodal features of RGB-D for recognition, and introduced a unidirectional complementary multimodal fusion method to mitigate the impact of negative depth information. Despite reaching an AP50 of 91.12%, the parameter count increment amounts to 17.8% of the original YOLOv7, which is not favorable for practical deployment in production. Concurrently, this approach results in the acquisition of depth information consuming more material and financial resources. Wang, Tao, et al. [6] et al. proposed an R-CNN based Mask-RCNN network for tea picking point detection. The network achieves an average accuracy of 93.95% and a recall of 92.48% and is robust under natural conditions. Within the realm of tea pest and disease target detection, a novel deep learning framework proposed by Hu, Gensheng, et al. [7] has been introduced to address the challenge of detecting tea wilt and discerning its severity. This innovative framework enhances the detection of tea pests and diseases, even in the presence of

fuzzy, occluded, and diminutive targets. Soeb, Md Janibul Alam, et al. [8], on the other hand, pre-processed five pest and disease datasets before feeding them into the YOLOv7 network and achieved 96.7% accuracy and 96.4% recall.

In order to solve the problems of poor visual characteristics and high noise of small objects in small object detection, this research constructed an end-to-end small object detection framework STF-YOLO. For small object detection of tea buds, Swin Transformer is innovatively introduced, using its local perception and global correlation to enhance the detection capabilities of tea buds. Furthermore, to enhance accuracy and speed, we incorporate modules such as Focus, Depthwise Convolution, SPPCSPC (Spatial Pyramid Pooling with Contextual Spatial Pyramid Convolution), and the C2 module into STF-YOLO. Our model successfully detects fresh tea buds measuring 2-3cm in length, even in complex background scenarios, while maintaining high accuracy. Evaluation of our dataset confirms the superior performance of the proposed model, surpassing other detectors with a precision of 91.5% and an mAP of 0.894, and the FPS reached 60.98. Compared with other current mainstream algorithms, its average accuracy improved by 5-20.22 percentage points, and the F1-score improved by 0.03-0.13 percentage points. The key contributions of this paper can be summarized as follows:

1. A dataset of tea bud images in natural environments was constructed. The dataset consists of 2898 original images, which were augmented to a total of 6242 images using data augmentation techniques.

2. This study introduces a novel tea bud detection model called STF-YOLO. Based on YOLOv8, we incorporate modules such as Swin Transformer, Focus, Depthwise Convolution, Spatial Pyramid Pooling with Contextual Spatial Pyramid Convolution (SPPCSPC), and the C2 module.

3. By utilizing the improved single-stage recognition technique, the recognition of tea buds is achieved, leading to improved accuracy in small object detection. Compared to other dark detectors and low-light enhancement models, our STF-YOLO achieves advanced results on the tea bud dataset.

4. This research explores the effectiveness of mainstream attention mechanisms for small object detection. The study involves the use of over ten different attention mechanism modules, including Res CBAM, CA, and SEAttention, for detecting tea buds.

## Related work

### Object detection

Object detection has always been one of the fundamental challenges in the field of computer vision. With the improvement in GPU computing power, deep learning-based object detection has gradually become mainstream. It can be mainly divided into two categories: one-stage and two-stage algorithms. The main difference lies in the fact that two-stage networks generate candidate regions after feature extraction and then perform classification or regression. Among the two-stage series, R-CNN, proposed by Girshick, Ross, et al. [9], can be considered a pioneering work in deep learning-based object detection. It utilizes the Selective Search algorithm to generate candidate bounding boxes and employs CNN for feature extraction. Although it significantly improves the mean average precision (mAP), it is slow in terms of detection speed and occupies a large amount of space. Based on this, they further proposed Fast R-CNN [10]. Under the same backbone network, the speed has increased by 9 times, and the mean average precision (mAP) on the PASCAL VOC 2012 dataset has improved to 65.7%.

Ren, Shaoqing, et al. [11] introduced the Region Proposal Network (RPN) as a replacement for Selective Search in generating candidate regions. This approach achieved an mAP of 70.4% on the PASCAL VOC 2012 dataset and a frame rate of 5 fps on GPUs.

Although two-stage methods achieve high accuracy, their detection speed is not high. Therefore, YOLO, proposed by Redmon, Joseph, et al. [12], greatly improved the speed to 45 fps. Although it may have more localization errors, it is less likely to predict incorrect object information in the background. The introduction of YOLO can be considered the beginning of a new era of deep learning algorithms known for their speed, and it further promotes the widespread application of deep learning algorithms. However, the progress of YOLO did not stop there. YOLOv2 [13] aimed to be better, faster, and more powerful. It not only achieves a trade-off between speed and accuracy but also introduces scale variability in input image sizes and expands the number of detectable object categories to over 9,000. YOLOv3 [14] made a series of small improvements, including changing the backbone network, classifier, and adding the SPP block. At the same time, the authors explicitly mentioned that previous versions of YOLO struggled with detecting small objects. However, YOLOv3 demonstrated a reversal of this trend. YOLOv3 performs better at detecting small objects compared to medium and large objects. Subsequent versions of the YOLO series have continued to improve, making YOLO more integrated, user-friendly, and deployable. Starting with YOLOX, proposed by Ge, Zheng, et al. [15], the YOLO series has entered the era of anchor-free detection, which has further enhanced its speed. YOLOv8 can be considered a fusion of the innovative ideas proposed in previous versions of YOLO. It is a target detection algorithm specifically designed for application deployment. Nowadays, many innovative networks are improving or adding modules to single-stage models. Roy, Arunabha M., and Jayabrata Bhaduri. [16] et al. proposed a real-time, high-performance damage detection model, DenseSPH-YOLOv5, based on deep learning techniques, which incorporates a CBAM module for extracting deep spatial features. It attached a spatial blending layer and a Swin-Transformer header to detect objects of different sizes and also reduces the computational complexity. Roy, Arunabha M., et al. [17] proposed the WilDect-YOLO network for real-time target detection in wildlife, which introduces a residual module in the CSPDarknet53 backbone to make the model powerful and discriminative deep spatial feature extraction and a DenseNet module to enable the model to retain key feature information. At the same time, SPP and PANet were introduced. And its mAP reached 96.89% in the wildlife dataset, with an F1 value of 97.87%.

## Object detection data augmentation

Despite continuous updates and improvements in deep learning-based object detection algorithms, the task of object detection is still hindered by practical issues. These include inconsistent dataset quality, insufficient quantity, incomplete category coverage, and complex backgrounds in natural environments. These issues hinder the faster and more effective application of object detection in various domains of production and daily life.

Image augmentation is a common technique used for data augmentation in computer vision tasks. In datasets captured in natural environments, complex backgrounds pose challenges due to factors such as lighting variations and object occlusions. Wu, Delin, et al. [18] pointed out that oil tea orchards have complex environments. The captured dataset can be affected by side lighting, background light, occlusions (both slight and severe), and object overlap. These factors can result in false positives or false negatives in object detection. To address this issue, they employed methods such as horizontal and vertical flipping, brightness augmentation, reduction, multi-angle rotation, and the addition of Gaussian noise and Mosaic data augmentation. Ultimately, they achieved an accuracy of 94.21% and a recall rate of

95.74% on the YOLOv7 network. DeVries, Terrance, and Graham W. Taylor [19], on the other hand, proposed a simple regularization technique called Cutout. This technique involves randomly masking out rectangular regions during training. This method is helpful in improving the robustness and overall performance of convolutional neural networks.

Another common technique for augmenting image data is image mixing. Zhang, Hongyi, et al. [20] proposed a method called "mixup," which involves randomly mixing two images using a mixing factor. Hendrycks, Dan, et al. [21] introduced a method called AugMix, which aims to enhance the robustness of data against positional variations during deployment.

There are many other data enhancement methods borrowed from data synthesis today, the most used of which is the GAN family of networks. Zhao, Qianxi, Liu Yang, and Nengchao Lyu. [22] used WGAN for data enhancement and combined it with a deep convolutional RNN network for real-time target detection, which resulted in significant improvements in both precision and recall. Ravikumar, R., et al. [23] fed preprocessed datasets with clear grey matter representations into cGAN to generate more training examples and used stacked CNN layers for feature extraction.

## Small object detection of crops

Small object detection, as a branch of object detection, has gradually gained attention. Initially, small object detection was primarily applied in domains such as face recognition, traffic sign detection, and pedestrian detection. Zhang, Yan, et al. [24] proposed a lightweight and detail-sensitive PAN for multi-scale industrial defect detection using YOLOv8 as a framework. It achieves mAPs of 80.4, 95.8%, and 76.3% under three publicly available datasets, namely, NEU-DET, PCB-DET, and GC10-DET, respectively. Cao, Xuan, et al. [25] improved the model based on Swin Transformer and YOLOv5, introduced CIOU to enhance the K-means clustering algorithm, and the modified CSPDarknet53 combined with Swin Transformer was used to extract more differentiated features, and CA was introduced into YOLOv5 for improving the performance of small object detection on remote sensing images. In the DOTA dataset, it achieves a mAP of 74.7%, which is an improvement of 8.9% compared to YOLOv5. Guo, Feng, et al. [26] proposed a model called Crack Transformer, which unifies Swin-Transformer as an encoder and decoder accompanied by an MLP layer, for automatic detection of long and complex road cracks. This study demonstrates the feasibility of using a Transformer-based network for road crack inspection in complex situations. Li, Feng, et al. [27] proposed a unified target detection and semantic segmentation framework, Mask-DINO, which extends DINO by adding a mask prediction branch that supports all image segmentation tasks (instance, panorama, and semantic). experiments show that this model has significant advantages over all current semantic segmentation models.

However, there is limited research on small object detection, specifically in the context of crops. This paper compiles previous studies on small object detection in crops, with a specific emphasis on tea leaves. Due to the small and densely distributed top buds of Chinese fir seedlings, existing recognition algorithms suffer from a high number of misjudgments. Yes, Zhangxi et al. [28] proposed a small object detection algorithm based on YOLOv5. To improve the model's ability to detect small objects, they added a micro-prediction head to the original detection head. They also incorporated a multi-attention mechanism module that combines CBAM (Convolutional Block Attention Module) and ECA (Efficient Channel Attention) into the model's architecture. Additionally, data augmentation and test-time augmentation methods were used. The obtained results were excellent and effectively addressed the challenges associated with detecting small objects.

To accurately identify and classify tender buds such as "one bud, one leaf" and "one bud, two leaves," Yu, Long, et al. [29] proposed an improved SS-YOLOX algorithm based on YOLOX. They introduced the SE attention mechanism and Soft NMS to enhance recognition capability and improve the recall rate. With a rich dataset, the mean average precision (mAP) achieved in this study was 86.3%. However, the authors made limited modifications to the network, and further improvements in recognizing small objects can be achieved by changing the IoU loss function and utilizing multimodal data. Shuai, Luyu, et al. [30] created a multimodal dataset that includes RGB images, depth maps, and infrared images. They trained the modified YOLOv5 network using this dataset and proposed two data-level multimodal image fusion methods as well as a feature-level multimodal image fusion method. The accuracy reached 85%, and the mean average precision at 50 (mAP@50) reached 82.7%. However, the depth maps and infrared images in this study contained some noise, blank areas, and limitations in equipment due to the lack of sufficient natural tea leaf images. As a result, the experimental results were suboptimal. Xue, Zhenyang, et al. [4] proposed the YOLO-Tea network, which introduced the ACmix and CBAM modules into the YOLOv5 network to enable more effective attention to small objects. However, this network did not perform well on all evaluation metrics, indicating the need for further research and improvements.

## Materials

The construction of the tea bud dataset involved three main steps: dataset collection, data preprocessing, and dataset generation. Initially, drones were employed to capture aerial photographs of the tea plantation, thereby gathering raw data for tea bud images. These operations resulted in an expansion of the initial image count from 322 to 6242. To ensure accurate classification and labeling of the tea bud images, the LabelImg tool was utilized. This step provided a reliable foundation for subsequent data analysis and model training. Finally, the processed data was transformed into a dataset in PASCAL-VOC2007 format, facilitating further research and applications. The construction process of such a dataset offers high-quality data resources, thereby providing robust support for research on tea bud images. The main processes are illustrated in Fig 1.

### Data collection

We employed unmanned aerial vehicles (UAVs) to capture video footage of Wanmu Tea Garden located in Hejiang Town, Yucheng District, Ya'an City, Sichuan Province, and have

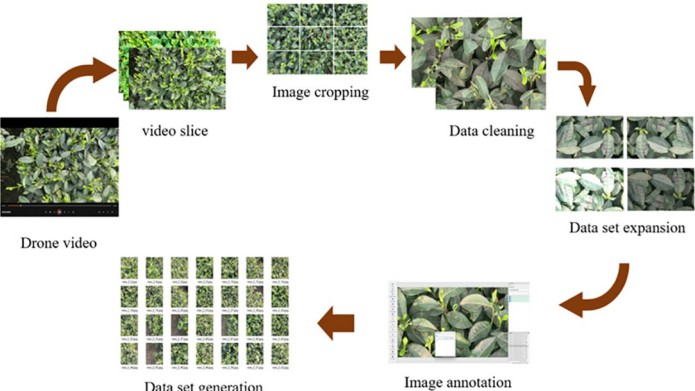

**Fig 1. Tea bud dataset production process.**

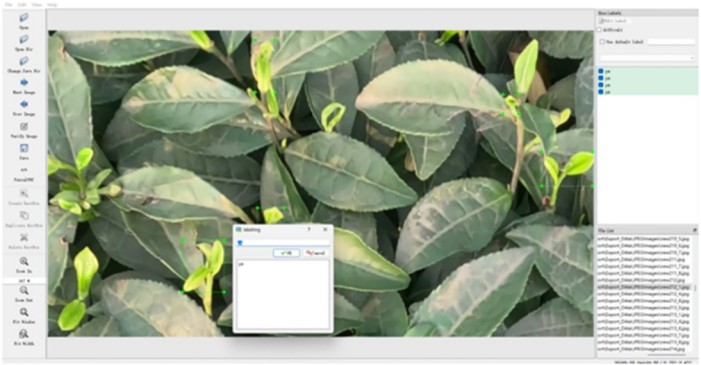

**Fig 2. LabelImg work labeling interface.**

obtained a filming permit. We specifically selected raw videos that showcased abundant tea buds for data collection. To guarantee the dataset's quality and applicability, our initial step was to establish the appropriate flight altitude for data collection. For this study, two distinct collection heights were chosen during a site visit: 15cm to 25cm and 90cm to 100cm from the tea tree. The height range of 90cm to 100cm was selected to enable a wider scanning of the tea plantation, allowing for a comprehensive assessment of the overall sprouting situation. On the other hand, the height range of 15cm to 25cm was chosen to ensure the clarity of the collected data and the visibility of the buds, thereby guaranteeing efficient data capture. If the height exceeds 100cm, the identification of tea buds will pose a challenge. Conversely, if the height falls below 15cm, the scanning of the tea plantation will become inefficient and impractical for production purposes. In the present investigation, a series of 10 videos depicting tea buds were recorded utilizing an Unmanned Aerial Vehicle (UAV). The total duration of these videos amounted to 15 minutes and 4 seconds, excluding the time taken for the UAV to take off and land. Consequently, the effective duration of the captured footage was determined to be 10 minutes and 33 seconds. Subsequently, the video files that were captured underwent sampling and slicing operations to generate the original dataset of images. A sampling frequency of one picture per 60 frames was utilized, leading to the generation of a dataset comprising 322 pictures. Data annotation is shown in Fig 2.

## Data cleaning

To improve the quality and usability of the tea bud image data, this study utilizes a data-cleaning methodology for data preprocessing. During the process of capturing images of tea shoots in tea plantations, it is common for the captured images to contain multiple duplications and blurring due to lens rotation and UAV movement. Consequently, the dataset of tea bud images utilized in this study requires data-cleaning procedures to ensure the integrity and dependability of the data.

(1) Eliminating blurry images: The image dataset of tea buds contains several blurry pictures, as depicted in Fig 3. These blurriness issues are attributed to the shaking of the UAV during the image capture process or the impact of wind. These indistinct images can have a detrimental impact on the training of machine learning models. They have the potential to convey misleading information and lack sufficient valid data for effective model training. Furthermore, they may misdirect the model's learning process by focusing on incorrect features. Hence, it is crucial to execute the process of eliminating blurry images from the dataset of tea bud images.

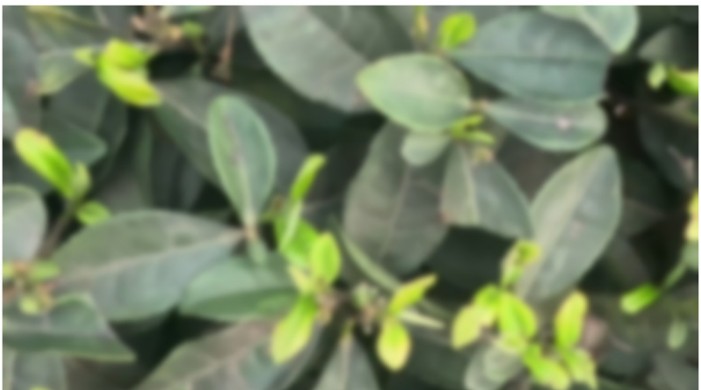

**Fig 3. Blurred image example.**

In the domain of computer vision, the evaluation of image sharpness frequently depends on the analysis of the image's gradient. The gradient is a measure of the magnitude of pixel variations, which offers valuable insights into the edges and textures present in an image. To assess the clarity of the images, the present study utilizes the Tenengrad gradient function for the computation of the image gradient. The assessment of image clarity is determined by the magnitude of the gradient value, with a higher gradient value indicating a higher level of image clarity. The process involves several steps. Firstly, the image is converted into a grayscale map. Then, the Tenengrad function is applied to calculate the gradient of the image. The resulting Tenengrad value is obtained. Next, the average value of all Tenengrad values is determined to be 56.77. Finally, to maintain the complexity of the data, a threshold slightly lower than the average value, specifically 44, is set. Any images with a Tenengrad value below this threshold are deleted [31]. The specific calculation is demonstrated in the following Eqs 6 and 7, where Gx and Gy represent the convolution values obtained from Sobel's edge detection operator for the horizontal and vertical modes of the pixel (x, y), respectively. The Sobel operator convolution template is defined by Eq 7, where I represent the original image data value.

$$\mathbf{D}(f) = \sum_{y}\sum_{x}|\sqrt{\mathbf{G}_x^2(x,y) + G_y^2(x,y)}| \tag{1}$$

$$G_x = \begin{pmatrix} 1 & 0 & -1 \\ -2 & 0 & 2 \\ -1 & 0 & 1 \end{pmatrix}^* \mathbf{I}, G_y = \begin{pmatrix} 1 & 2 & 1 \\ 0 & 0 & 0 \\ -1 & -2 & -1 \end{pmatrix}^* \mathbf{I} \tag{2}$$

(2) Elimination of redundant images: Due to the significant resemblance between consecutive images and the abundance of similar images in the dataset, the potential utilization of these images multiple times during model training can lead to overfitting of the model. Additionally, the existence of comparable images can impede the model's capacity for generalization, thereby posing difficulties in its ability to adapt to novel data. Consequently, the removal of similar images can greatly improve the accuracy and stability of the model, particularly for datasets containing tea bud images.

In this study, the Structural Similarity Measure (SSIM) was chosen to calculate the similarity of the images and remove the similar ones. The Structural Similarity Index (SSIM) considers three key aspects of an image: brightness, contrast, and structure. It calculates a metric

**Table 1. Data processing.**

| Number of original datasets | Number of fuzzyimages deleted | Number of similar images deleted | Number of manual deletions | the final dataset |
|---|---|---|---|---|
| 2898 | 822 | 234 | 611 | 1231 |

value for structural similarity ranging from 0 to 1, where higher values indicate greater similarity between the two images [32]. The specific steps are as follows: First, convert the two images into grayscale images. Then, divide the grayscale images into blocks and calculate the brightness, contrast, and structural similarity of each block. Multiply the similarity of the blocks and take the mean value as the structural similarity measure of the two images. The final similarity is the average structural similarity measure. Finally, set the threshold to 0.8 and use the structural similarity obtained by the two images if it is greater than 0.8. In that case, one of them is deleted. The specific calculation formula for SSIM is shown in Eq 3, where x and y represent the two images being compared. $\mu_x$ and $\mu_y$ represent the mean brightness of images x and y, respectively. $\sigma_x$ and $\sigma_y$ represent the standard deviations of images x and y, respectively, while σxy represents the covariance between images x and y. This formula quantifies the level of structural similarity between the two images.

$$SSIM(x, y) = \frac{(2\mu_x\mu_y + c_1)(2\sigma_{xy} + c_2)}{(\mu_x^2 + \mu_y^2 + c_1)(\sigma_x^2 + \sigma_y^2 + c_2)} \tag{3}$$

After applying the aforementioned data cleaning steps, the obtained image information is presented in Table 1-2. The original dataset consisted of a total of 2,898 images. After removing 822 blurred images and 234 similar images, a final dataset of 1,842 images was obtained.

Manual deletion is still required to improve the quality and trustworthiness of the data, which is necessary for supporting subsequent modeling and training of deep learning algorithms. After manually removing 611 image data entries, a final set of 1,231 image data entries was obtained, as presented in Table 1 below.

## Data preprocessing

In this study, tea bud images were captured using a UAV with a resolution of 2.7K (2074*1520 pixels). During the training of deep learning models, it is common to resize the images. When scaling the tea bud images, the pixel values tend to be biased towards smaller scales. This bias can have an impact on the texture features of the shoots. In addition, the small size of the buds in the enlarged image is also prone to labeling errors, which in turn affects the accuracy of target detection results. Thus, to mitigate the impact of scaled images on the training results and preserve clear information about the characteristics of the tea buds, the captured images were cropped in this study. A total of 2898 images were obtained by cropping the images into 9 parts, respectively, according to an aspect ratio of 3:3. The images were cropped to a uniform resolution of 901*506 pixels. See Fig 4 for details.

Due to the limitations in capturing images of tea buds, such as the limited quantity and issues like color imbalance and uneven distribution of samples, it is crucial to address these challenges and improve the accuracy and recall rate of recognizing tea bud targets in natural environments using object detection algorithms. To address these issues, it is necessary to expand the tea bud image dataset. Specifically, dataset augmentation is used to increase the diversity and scale of the dataset, thereby enhancing the model's generalization capability, robustness, and mitigating overfitting. To achieve dataset augmentation, various

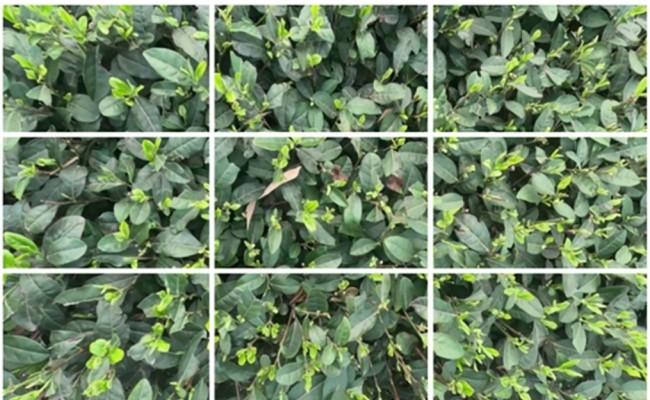

**Fig 4. Cropped image.**

transformations such as random stretching, brightness adjustment, and mirroring were applied to the original images. These operations aim to increase the size and diversity of the dataset.

Among these augmentation techniques, image stretching is an effective method for expanding the dataset by altering the aspect ratio of the images. This helps to increase the diversity of the dataset. It can be performed in two ways: vertical stretching and horizontal stretching. Considering the significant aspect ratio of the original images, we chose to use vertical stretching to enhance the original dataset. Vertical stretching involves adjusting the height of the images, either stretching or compressing them, which results in taller or shorter images and modifies the aspect ratio accordingly. This approach helps maintain the integrity of the image information while introducing variations in the dataset.For example, when observing the image after vertical stretching (as shown in Fig 5(b)), it is evident that the aspect ratio has changed significantly compared to the original image (Fig 5(a)). This image-stretching operation helps increase the diversity of the dataset and enhances the recognition capability of tea bud targets.

In order to expand the dataset and increase its diversity, the brightness values of the images were adjusted. By flipping the images along the "center axis," new training samples can be generated, thereby increasing the number of samples in the dataset. Image mirroring can be performed in three ways: horizontal, vertical, and diagonal mirroring.

The original 1231 images were expanded using stretching, mirroring, and brightness adjustment methods, resulting in 6242 images. As shown in Table 2 below, the original images

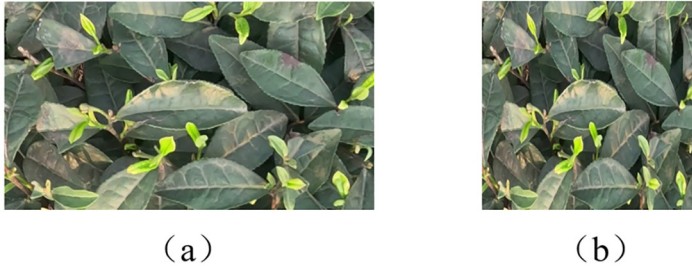

（a）                （b）

**Fig 5. a) original image (b) Vertically stretched image.**

**Table 2. Data set expansion.**

| Means of expansion | amount | percentage |
|---|---|---|
| original images | 1231 | 19.80% |
| Stretcheds images | 697 | 11.16% |
| Brightness-adjusted images | 1346 | 21.56% |
| Mirror images | 2963 | 47.46% |

totaled 1236, accounting for 19.80%. The stretched images totaled 697, accounting for 11.16%. The total number of images after brightness adjustment is 1,346, accounting for 21.56%. The total number of mirrored images is 2963, accounting for 47.46 percent.

## Methods

In this section, we will present in detail the approach to solving the problems of background complexity and small object detection. In order to address the difficulty of detecting and classifying tea buds in the field of tea target detection, we propose an improved method called STF-YOLO, which combines YOLOv8 with Swin Transformer to enhance the feature fusion capability of the model and introduces Focus, Depthwise Convolution [33], Spatial Pooling Pyramid [34], and the C2 module, which enable the model to achieve good results in the task of bud detection for small objects with complex backgrounds. Fig 6 is the overall structure of our proposed network model.

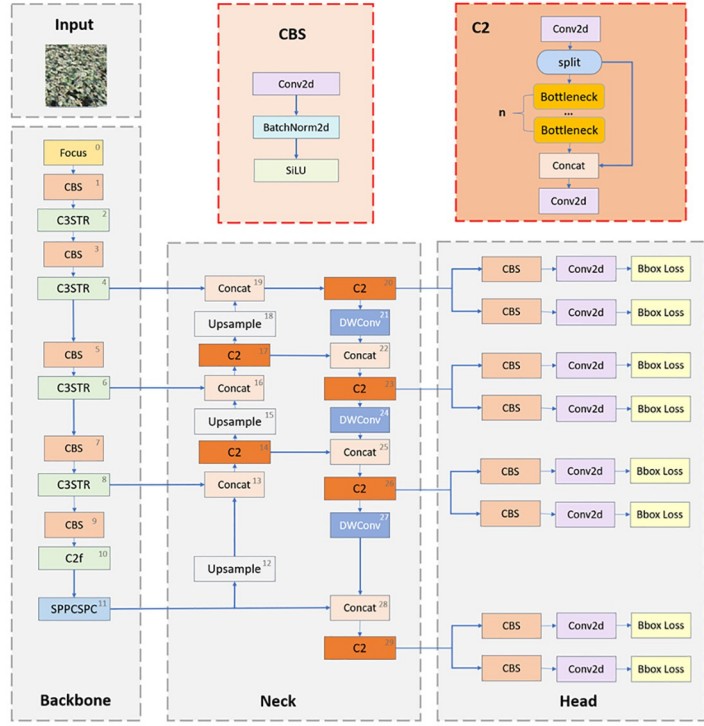

**Fig 6. STF-YOLO structure diagram.**

## Overview of the YOLOv8

YOLOv8 is a state-of-the-art object detection algorithm that combines the modified backbone network of YOLOv5 with the C2f module to introduce an anchor-free model that uses decoupled heads for independent processing of objectivity, classification, and regression tasks. The model employs a sigmoid function for objectivity scores and a softmax function for category probabilities. YOLOv8 uses CIoU and DFL loss functions for bounding box loss computation and binary cross-entropy for classification loss computation, resulting in improved performance, especially for detecting smaller objects. In addition, YOLOv8 provides a semantic segmentation model called YOLOv8-Seg, which achieves state-of-the-art results in a variety of benchmarks while maintaining high speed and efficiency.

Detecting tiny objects is a very challenging problem because the size of a tiny object contains only a few pixels, and its features are often extracted from shallow features. Yolov8 [35] has made improvements in detecting small-sized targets, but it is still not able to adequately capture the contextual information around the target. Features of small objects are usually extracted from shallow features, which may lack sufficient semantic information to provide a rich contextual background. The lack of contextual information may cause YOLOv8 to encounter difficulties in detecting and localizing small objects.

For small object detection, several researchers have proposed corresponding improvement methods. Introducing multi-scale feature extraction is a common strategy for improving small object detection performance. Traditional networks [36] use a single-scale feature map for detection, but this may not capture detailed information about different object sizes. To solve this problem, researchers have proposed a multi-scale feature extraction approach. This approach allows for the simultaneous processing of information from objects of different sizes by introducing multiple feature maps of different scales into the network. Features are usually extracted at different layers of the network, and these features are fused to improve the detection of small objects. Introducing an attention mechanism is another way to improve the detection of small objects. Representation is improved by using the attention mechanism: focusing on important features and suppressing unnecessary ones. The attention mechanism can weigh different regions of the feature map according to the importance of the object, making the network pay more attention to the representation of small objects. [37] This improves the attention of small objects and enhances their feature representation. Researchers have proposed various variants of attention mechanisms such as channel attention, spatial attention, and multiscale attention. These methods can adjust the attention weights on different dimensions of the feature map, thus enhancing the detection performance of small objects. Top-down information, context and feedback also play important roles in object detection, and combining contextual information about the object is also an important method to improve small object detection [38].

To further improve YOLOv8 and enhance its performance in small object detection, we address its lack of contextual and semantic information by introducing a Swin transformer to increase feature fusion. Swin Transformer establishes global dependencies in different regions of the feature map through the self-attention mechanism, thus effectively capturing contextual information. Swin Transformer [39] also introduces a windowed attention mechanism that reduces computational complexity by dividing the feature map into different windows while maintaining the global field of view range. In addition, we added the Focus with Deep Convolution DWconv module, which allows the network to better capture the vast contextual information and thus better understand the structure and features of the background, which helps to accurately segment and localize small object tea buds. The introduction of the spatial pooling pyramid SPPCSPC as well as the C2 module helps the model to perceive and analyze small

object buds at different scales effectively and improves the model's ability to detect multi-scale objects. After experimental testing, the network possesses a better performance effect in the small object bud detection task with a complex background.

## Modules

Table 3 is the detailed parameters of our improved STF-YOLO model to show the improved situation more clearly. The table consists of six columns, each representing a different aspect of the model. The columns are as follows:

- The first column is the number of layers.

- from: This column specifies the source of the input for each layer. -1 means that the input comes from the previous layer.

- n: This column specifies the number of times that a certain layer type is repeated.

**Table 3. Model parameters.**

|  | from | n | params | module | arguments |
|---|---|---|---|---|---|
| 0 | -1 | 1 | 8800 | ultralytics.nn.modules.conv.Focus | [3, 80, 3] |
| 1 | -1 | 1 | 115520 | ultralytics.nn.modules.conv.Conv | [80, 160, 3, 2] |
| 2 | -1 | 3 | 286710 | ultralytics.nn.modules.block.C3STR | [160, 160, 3] |
| 3 | -1 | 1 | 461440 | ultralytics.nn.modules.conv.Conv | [160, 320, 3, 2] |
| 4 | -1 | 6 | 2068510 | ultralytics.nn.modules.block.C3STR | [320, 320, 6] |
| 5 | -1 | 1 | 1844480 | ultralytics.nn.modules.conv.Conv | [320, 640, 3, 2] |
| 6 | -1 | 6 | 8233020 | ultralytics.nn.modules.block.C3STR | [640, 640, 6] |
| 7 | -1 | 1 | 5531520 | ultralytics.nn.modules.conv.Conv | [640, 960, 3, 2] |
| 8 | -1 | 3 | 10170285 | ultralytics.nn.modules.block.C3STR | [960, 960, 3] |
| 9 | -1 | 1 | 11061760 | ultralytics.nn.modules.conv.Conv | [960, 1280, 3, 2] |
| 10 | -1 | 3 | 27865600 | ultralytics.nn.modules.block.C2f | [1280, 1280, 3, True] |
| 11 | -1 | 1 | 44254720 | ultralytics.nn.modules.block.SPPCSPC | [1280, 1280, 5] |
| 12 | -1 | 1 | 0 | torch.nn.modules.upsampling.Upsample | [None, 2, 'nearest'] |
| 13 | [-1, 8] | 1 | 0 | ultralytics.nn.modules.conv.Concat | [1] |
| 14 | -1 | 3 | 15523200 | ultralytics.nn.modules.block.C2 | [2240, 960, 3, False] |
| 15 | -1 | 1 | 0 | torch.nn.modules.upsampling.Upsample | [None, 2, 'nearest'] |
| 16 | [-1, 6] | 1 | 0 | ultralytics.nn.modules.conv.Concat | [1] |
| 17 | -1 | 3 | 6969600 | ultralytics.nn.modules.block.C2 | [1600, 640, 3, False] |
| 18 | -1 | 1 | 0 | torch.nn.modules.upsampling.Upsample | [None, 2, 'nearest'] |
| 19 | [-1, 4] | 1 | 0 | ultralytics.nn.modules.conv.Concat | [1] |
| 20 | -1 | 3 | 1795200 | ultralytics.nn.modules.block.C2 | [960, 320, 3, False] |
| 21 | -1 | 1 | 3520 | ultralytics.nn.modules.conv.DWConv | [320, 320, 3, 2] |
| 22 | [-1, 17] | 1 | 0 | ultralytics.nn.modules.conv.Concat | [1] |
| 23 | -1 | 3 | 6560000 | ultralytics.nn.modules.block.C2 | [960, 640, 3, False] |
| 24 | -1 | 1 | 7040 | ultralytics.nn.modules.conv.DWConv | [640, 640, 3, 2] |
| 25 | [-1, 14] | 1 | 0 | ultralytics.nn.modules.conv.Concat | [1] |
| 26 | -1 | 3 | 14908800 | ultralytics.nn.modules.block.C2 | [1600, 960, 3, False] |
| 27 | -1 | 1 | 10560 | ultralytics.nn.modules.conv.DWConv | [960, 960, 3, 2] |
| 28 | [-1, 11] | 1 | 0 | ultralytics.nn.modules.conv.Concat | [1] |
| 29 | -1 | 3 | 26636800 | ultralytics.nn.modules.block.C2 | [2240, 1280, 3, False] |
| 30 | [20, 23, 26, 29] | 1 | 15465236 | ultralytics.nn.modules.head.Detect | [1, [320, 640, 960, 1280]] |

- params: This column provides the number of parameters in each layer.

- module: This column specifies the type of the layer. For example, ultralytics.nn.modules. conv.Conv is a convolutional layer, ultralytics.nn.modules.block.C2f is a specific module that may contain multiple layers, ultralytics.nn.modules.block.SPPF is a spatial pyramid pooling layer.

- arguments: This column provides the arguments that are passed to the layer constructor.

**C3STR: Swin transformer module.** In multiple convolution operations. Most of the object features that small objects in the image should have will be lost continuously, so we borrowed the idea of Swin Trasformer in the operation of feature fusion. We introduced the C3STR module into the YOLOv8 algorithm, and its core idea is to establish global dependencies on different spatial locations of the feature map through the self-attention mechanism and enhance the semantic information and feature representation of small objects with the help of the window self-attention module. It is able to perform adaptive feature interaction for each location in the feature graph to capture the contextual information around the object. The module contains pairs of Window Multi-Headed Self-Attention Modules, Sliding Window Multi-Headed Self-Attention Modules, and Multi-Layer Perceptron Mechanisms, and each of them is internally connected using residual connections. The C3STR structure diagram is shown in Fig 7. The computational procedure of the multi-head self-attention mechanism is as follows:

$$\text{Attention}(Q, K, V) = \text{SoftMax}(QK^T/\sqrt{d} + B)V \qquad (4)$$

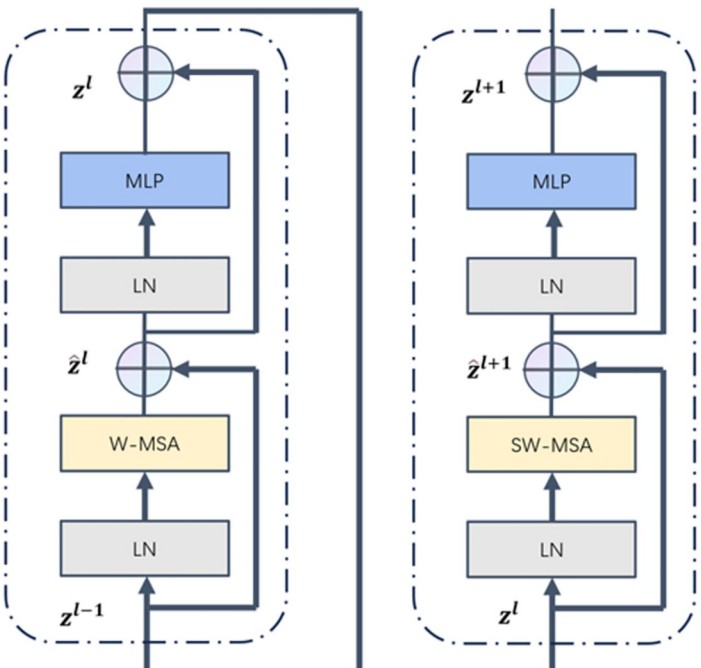

**Fig 7. C3STR structure diagram.**

In the above equation, Attention denotes the attention; SoftMax denotes the normalized exponential function; Q, K, and V are the query, key, and value matrices, respectively; d is the number of channels of the input feature map; and B is the relative positional bias. Introducing B can have an obvious enhancement effect. Compared with the traditional multi-head self-attention module in Transformer, the C3STR module controls the computational region in each window by dividing local windows to achieve cross-window information interaction, which reduces the computational complexity and network computation.

**Focus, DWconv, SPPCSPC and C2 module.** The SPPCSPC module is a deep learning module that combines spatial pyramid pooling and channel spatial pyramid convolution, the core idea of which is to efficiently capture feature information at different scales by applying spatial pyramid pooling and channel spatial pyramid convolution operations to the input feature map. The module will partition the input feature map into multiple regions of different scales, perform pooling operations on each region, and finally stitch the pooling results of different scales together. This captures contextual information at different scales and enables the model to better model objects of different sizes. The SPPCSPC structure diagram is shown in Fig 8.

The Dwconv (Depthwise Convolution) module is utilized for convolution operations in image processing and computer vision tasks. It enables separate convolution of each channel in the input feature map, meaning that each channel has its convolution kernel. This operation reduces the number of parameters and the amount of computation, thereby significantly reducing the complexity and storage requirements of the model while maintaining relatively high performance. This approach enables the model to effectively learn the spatial and channel information in the input feature map, enhancing the model's expressive and perceptual capabilities. The DWconv schematic is shown in Fig 9. The Focus module is an attention mechanism module designed for the task of object detection. It aims to improve the model's focus on important feature regions by applying a lightweight convolution operation to the input feature map. This operation partitions the feature map into multiple smaller sub-regions. These sub-regions are then weighted and fused by learning the resulting weights to produce the final attention feature map. This operation allows the model to efficiently focus on important feature regions. The Focus schematic is shown in Fig 10. The C2 module extracts high-level semantic features and enhances their expressiveness, thereby improving the performance of computer vision tasks. It is superior to the C2 module in terms of memory footprint and inference speed.

**Wiou loss function.** Addressing the issue of insufficient convergence observed in the YOLOv8 algorithm when trained on the tea bud dataset, this research paper introduces a novel approach by suggesting the incorporation of a loss function derived from Wise IOU to optimize the localization loss function of the YOLOv8 algorithm. This approach aims to enhance the convergence and generalization capabilities of the YOLOv8 model [39] by modifying the focus of the localization loss. It prioritizes the Intersection over Union (IOU) of predicted and real frames, leading to improved performance.

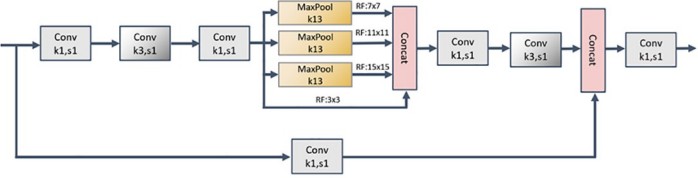

**Fig 8. SPPCSPC structure diagram.**

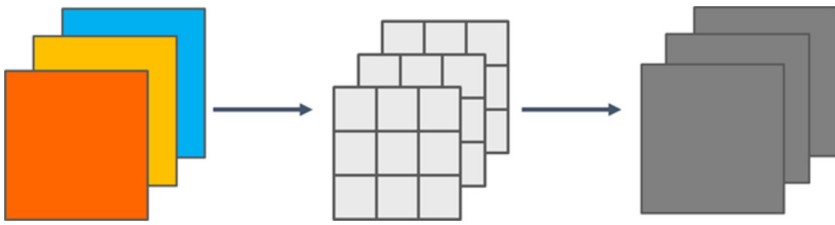

**Fig 9. DWconv schematic.**

By conducting an analysis of the loss function employed in the YOLO series of algorithms, this study reveals that the localization loss function commonly employs a mean-square error function. This function solely utilizes the coordinates and width-height information of the predicted frame and the actual frame for regression. Consequently, this approach is identified as one of the primary contributors to the instability observed during model training. In relation to the inherent characteristics of object detection, this study necessitates the identification of the predicted frame that exhibits the highest degree of overlap with the actual target frame. This criterion aligns most effectively with the objectives of object detection [40]. Therefore, this paper proposes a shift in the localization loss function from measuring the distance between the predicted frame and the real frame to evaluating the Intersection over Union (IOU) between them. This modification aims to better fulfill the fundamental requirement of object detection. Given the presence of low-quality examples in training data, it is important to consider the impact of geometric factors, such as distance and aspect ratio, on the penalty associated with these examples. These factors can exacerbate the negative effects of low-quality examples, ultimately leading to a reduction in the generalization performance of the model. Traditional methods for calculating IOU only take into account the proportion of intersections and concatenations between detection frames without considering their positional relationships within the image. A well-designed loss function should mitigate the impact of geometric factors when there is an overlap between the anchored frames and the aspect ratio. The impact of geometric factors on the penalty should be reduced in cases where the anchored frames align closely with the object frames. Additionally, minimizing intervention during training will lead to improved generalization of the model. To enhance the precision of measuring the coverage of the detection frame, some researchers have proposed a method called "Wise IOU" (WIOU) [39]. Based on this, the construction of distance attention relies on the utilization of a distance metric. In the present study, the WIOUv1 model is selected, which incorporates a

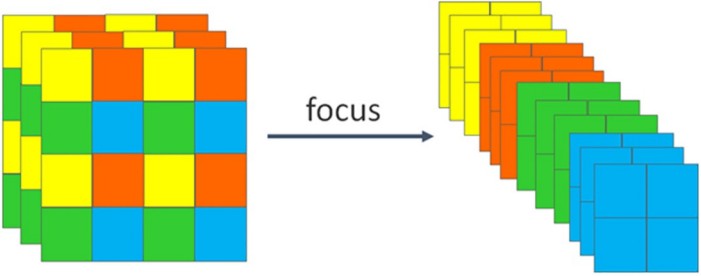

**Fig 10. Focus schematic.**

two-layer attention mechanism. The calculation formula for this model is presented below:

$$L_{\mathrm{WIOU\ v1}} = R_{\mathrm{WIoU}}\ L_{\mathrm{IOU}} \tag{5}$$

$$R_{\mathrm{WIOU}} = \exp\left(\frac{(x - x_{gt})^2 + (y - y_{gt})^2}{(W_g^2 + H_g^2)^*}\right) \tag{6}$$

$$L_{\mathrm{IOU}} = 1 - IOU = 1 - \frac{W_i H_i}{S_u} \tag{7}$$

The formula, $R_{\mathrm{WIOU}} \in [1, e)$, which significantly amplifies $L_{\mathrm{IOU}}$ for normal quality anchor frames, $L_{\mathrm{IOU}} \in [0, 1]$, which significantly reduces $R_{\mathrm{mov}}$ for high-quality anchor frames and decreases its emphasis on the centroid distance when anchor frames overlap well with the object frame. To mitigate the convergence hindrance caused by gradients in $R_{\mathrm{mov}}$, the computational map separates $W_g$ and $H_g$ (indicated by superscript *). In this paper, the authors propose a method that effectively addresses the factors that impede convergence. The introduced method does not introduce any new metrics. Instead, it utilizes the Intersection over Union (IOU) to quantify the level of overlap between the prediction frame and the real frame in the object detection task. The overlap region, depicted in Fig 11, is measured by the IOU metric and has an area denoted as

$$S_u = wh + w_{gt}h_{gt} - W_i H_i \tag{8}$$

## Indicators for model evaluation

In the domain of deep learning, the efficacy of network models is commonly assessed through the utilization of mean Average Precision (mAP) and Recall (R) metrics [39]. Many additional concepts are implicated in the computation of these two metrics, including Intersection over

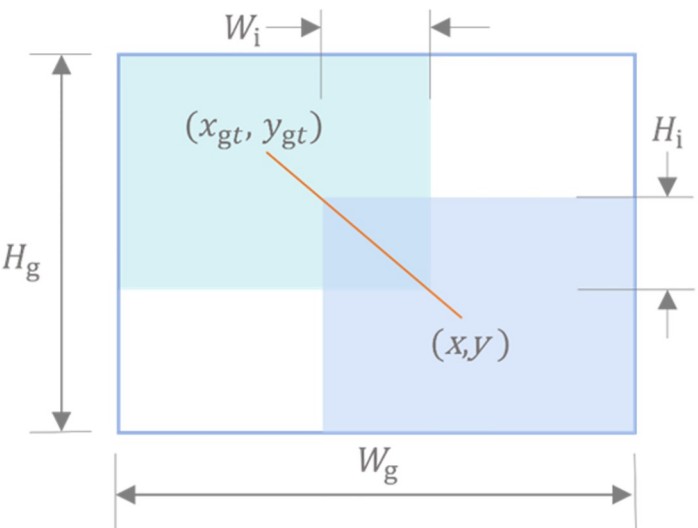

**Fig 11. The overlapping area between the prediction box and the real box.**

Union (IOU), Precision (P), and Average Precision (AP). Each of the descriptions is provided below.

(1) IOU The intersection and concatenation ratio is a quantitative measure employed to assess the level of overlap in the outcomes of an object detection algorithm. The calculation involves comparing the intersection and concatenation of the detection result's area with the area of the actual label. The intersection-merge ratio can be defined as the quotient of the intersection area and the concatenation area. The accuracy of the detection result increases as the value of the IOU calculation formula, as shown in Eq 9:

$$IOU = \frac{A \cap B}{A \cup B} \tag{9}$$

In the aforementioned equation, the variable A denotes the prediction frame, while the variable B represents the real frame. In the context of deep learning based object detection, it is common practice to define a large threshold value and a small threshold value. When the Intersection over Union (IOU) metric exceeds the large threshold value, the object recognition is deemed correct. Conversely, if the IOU falls below the small threshold value, the recognition is considered incorrect. In cases where the IOU value lies between the two thresholds, the recognition result is discarded.

(2) Precision and recall In the domain of object detection, the precision rate refers to the proportion of accurately detected objects in relation to the overall number of objects detected. The precision of the algorithm is inversely proportional to its false positive rate. The recall is defined as the proportion of correctly detected objects to the total number of positive samples. A higher recall signifies that the algorithm demonstrates improved capability in detecting all true positive samples. Eq 10 represents the precision, while Eq 11 represents the recall.

$$P = \frac{TP}{TP + FP} \tag{10}$$

$$R = \frac{TP}{TP + FN} \tag{11}$$

In the context of object detection, TP (True Positive) represents the count of correctly predicted shoot bounding boxes, FP (False Positive) represents the count of incorrectly predicted shoot bounding boxes, and FN (False Negative) represents the count of missed shoot bounding boxes. Thus, P represents the ratio of accurate predictions to all predictions, while R represents the ratio of accurate predictions to all true objects. It is important to note that both P and R have values ranging from 0 to 1.

(3) AP and mAP (Average Precision and mean Average Precision) The average precision (AP) is calculated as the mean precision at various recall points, representing the area under the precision-recall (P-R) curve. The mean Average Precision (mAP) is calculated as the average of the Average Precision (AP) values for all categories N. The range of mAP values is also between 0 and 1, and this can be mathematically represented by Eqs 12 and 13. In this thesis, the terms AP (Average Precision) and mAP (mean Average Precision) are equivalent due to

the nature of the task at hand, which involves the identification of tea buds, a single category.

$$AP = \prod_0^1 P(r)dr \tag{12}$$

$$mAP = \frac{1}{N}\sum AP \tag{13}$$

(4) FPS FPS (Frames Per Second) is the number of frames per second that can be processed by a computer when processing an image and is an important measure of the efficiency and speed of computer vision algorithms. In object detection evaluation, FPS can be used to measure the processing speed of an object detection algorithm, i.e., the number of objects that can be detected per second. A higher FPS value means that the algorithm is able to process more images and give a faster response in a short time.

## Experimental environment and parameter adjustment

The operating system utilized in this study was Windows 10, and PyTorch served as the framework for the development of deep learning models. Specific information regarding the experimental setting is provided in Table 4. During the training phase, optimization was performed using the stochastic gradient descent (SGD) algorithm. The SGD algorithm utilized an initial learning rate of 0.01, a momentum factor of 0.937, and a weight decay factor of 0.0005. The input image was normalized to a size of 640 × 640, the batch size was set to 8, and the training was conducted over 300 epochs.

## Results

### Overall accuracy comparison of network models

To assess the effectiveness of STF-YOLO, we conducted extensive experiments using the tea sprouts dataset. Our evaluation involved a comprehensive analysis and comparison of our enhanced model with various well-established detection models, with a particular emphasis on precision, recall, and mAP metrics. A line chart (Fig 12) was also generated to visualize and compare the mAP@0.5 curves of these models.

Our findings revealed that our improved STF-YOLO model achieved a higher mAP@0.5, surpassing the state-of-the-art GOLD-YOLO model by an impressive margin of 3.02 percentage points, as indicated in Table 5. Notably, STF-YOLO exhibited significant improvements across all target detection metrics, outperforming previous YOLO models. Specifically, the STF-YOLO algorithm showcased a precision of 91.5%, a recall rate of 77.6%, an mAP of 89.4% at 0.5 IoU, an mAP of 71% at 0.5:0.95 IoU, and an F1 score of 84%. These remarkable results demonstrate the substantial enhancement in detection accuracy achieved by the STF-YOLO

**Table 4. Experimental environment configuration.**

| Category | Configuration |
| --- | --- |
| CPU | Intel(R) Core(TM) i7-12700KF@3.60 GHz |
| GPU | GeForce RTX 3060 |
| System enviroment | Windows10 |
| Framework | Pytorch 2.0.0 |
| Programming voice | Pytorch 3.8 |

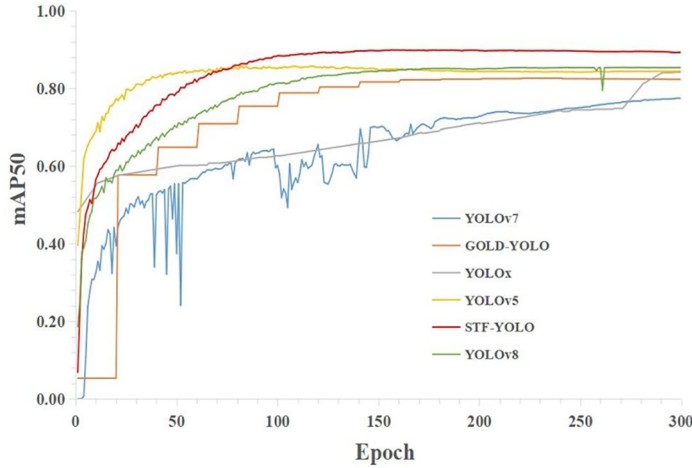

**Fig 12. mAP@0.5 line charts for different models.**

algorithm. Consequently, STF-YOLO proves to be highly suitable for the accurate detection of small tea bud objects, offering promising applications in the field.

Params refers to the number of parameters, which refers to the total number of parameters that need to be trained in model training. Used to measure the size of the model (i.e. computational space complexity). However, it is worth noting that the param of yolov8s is 11.2M, while STF-YOLO is 199M. While the accuracy is improved, the model becomes more complex and the amount of calculation is greatly increased.

## Ablation study

**Improvement in WIOU.** The original loss function calculation method of YOLOv8 was optimized to wIoU in this study, and comparative experiments were conducted. The experimental results are presented in the Table 6. It can be observed that the model's recall rate improved by 0.5 percentage points, precision improved by nearly 1.97 percentage points, mAP at 0.5 IoU increased by 2.3 percentage points, and the overall evaluation metric, F1 score, improved by 0.01.

**Effectiveness of mainstream attention mechanisms.** Additionally, we also tested the effectiveness of other mainstream attention mechanisms for small object detection, as shown in Table 7. Due to the unsatisfactory performance of attention mechanisms when incorporated into our model, we only analyze the impact of adding attention mechanisms to YOLOv8 for detecting small tea buds.

**Table 5. Comparison of different model algorithms.**

| Model | Precision(%) | Recall(%) | mAP@0.5(%) | mAP@0.5:0.95(%) | F1 Score |
|---|---|---|---|---|---|
| YOLOv7 | 71.28 | 71.18 | 77.42 | 49.11 | 0.71 |
| GOLD-YOLO | - | - | 82.28 | 59.50 | - |
| YOLO x | - | - | 84.10 | 59.20 | - |
| YOLOv5 | 88.20 | 75.89 | 84.32 | 67.65 | 0.82 |
| YOLOv8 | 86.5 | 76 | 85.3 | 65.7 | 0.81 |
| STF-YOLO | **91.5** | **77.6** | **89.4** | **71** | **0.84** |

**Table 6. Experimental results of YOLOv8+WIOU.**

| Model | Precision(%) | Recall(%) | mAP@0.5(%) | mAP@0.5:0.95(%) | F1 Score |
|---|---|---|---|---|---|
| YOLOv8 | 86.5 | 76 | 85.3 | 65.7 | 0.81 |
| YOLOv8+WIOU | **88.2** | **76.5** | **87.6** | **67** | **0.82** |

During the evaluation, it was observed that YOLOv8s had the slowest detection speed. However, after incorporating attention mechanisms, the detection speed improved significantly. The SEA (SEAttention) model exhibited the highest detection speed, reaching 526.32. The ESE (EffectiveSE) model achieved high performance across all metrics. The Shuffle (ShuffleAttention) model achieved the highest precision, reaching 0.897. In terms of F1 score, all models performed similarly. Overall, the impact of attention mechanisms on small tea bud object detection was not significant.

**Effectiveness of components in STF-YOLO.** To assess the effectiveness of each component in STF-YOLO, we conducted ablation experiments on the Focus, C2, DW (Depthwise Convolution), STF (Swin Transformer), and SPPCSPC modules. Performance evaluation was carried out using metrics such as Precision (%), Recall (%), mAP@0.5 (%), F1 Score, and FPS. The results are presented in Table 8.

Incorporating the STF module led to a slight decrease in FPS from 42.74 to 38.02, but resulted in an improvement in mAP from 88.8% to 89.2%. In order to enhance detection speed while maintaining accuracy, we replaced the SPPF module with SPPCSPC. As a result, precision slightly decreased from 92% to 91.5%, but the mAP increased by 0.2 percentage points. Notably, the detection speed significantly improved from 38.02 to 60.98 FPS, representing an approximate 60% increase compared to using the SPPF model. The detection effect of the model is shown in Fig 13.

## Discussion

In the context of tea bud detection, several challenges need to be addressed. Firstly, the small size of tea buds poses difficulties for feature representation and extraction. Secondly, the dense distribution and resulting occlusions make detection more difficult. Thirdly, complex lighting conditions affect visibility. Lastly, the morphological similarity between buds and leaves makes distinguishing them challenging.

Many studies on tea bud detection have incorporated attention mechanisms into their models, such as SS-YOLOX proposed by Yu et al. [45]. However, in our experiments, we tested various attention mechanisms in Table 7 and found that they did not significantly improve the detection performance, and there was no clear difference among them. Compared to YOLOv8, our STF-YOLO demonstrates better detection accuracy, recall, and mAP. However, this comes at the cost of a slightly slower detection speed, which may limit deployment on

**Table 7. Effectiveness of attention mechanisms.**

| Model | Precision(%) | Recall(%) | mAP@0.5(%) | mAP@0.5:0.95(%) | F1 Score | FPS |
|---|---|---|---|---|---|---|
| YOLOv8s | 86.5 | 76 | 85.3 | 62.2 | 0.81 | 277.8 |
| ESE [41] | 87.5 | **77.4** | **88.1** | **68.2** | **0.82** | 476.19 |
| Shuffle [42] | **89.7** | 74.8 | 87 | 67.1 | **0.82** | 500.0 |
| GE [43] | 88.9 | 76.3 | 87.5 | 67.5 | **0.82** | 500.0 |
| SEA [44] | 88 | 76.9 | 87.7 | 67.1 | **0.82** | **526.32** |
| Res_CBAM [37] | 88 | 76.9 | 87.7 | 67.5 | **0.82** | 500.0 |

**Table 8. Effectiveness of modules(Ablation experiment results).**

| Focus | C2 | DW | STF | SPPCSPC | Precision(%) | Recall(%) | mAP@0.5(%) | F1 Score | FPS |
|-------|----|----|-----|---------|--------------|-----------|------------|----------|-----|
| - | - | - | - | - | 88.2 | 76.5 | 87.6 | 0.82 | 277.8 |
| ✓ | - | - | - | - | 89.4 | 76.9 | 88.1 | 0.83 | 250.0 |
| ✓ | ✓ | - | - | - | 89.5 | 76.5 | 87.7 | 0.83 | 256.4 |
| ✓ | ✓ | ✓ | - | - | 92.6 | 76.7 | 88.8 | 0.84 | 42.74 |
| ✓ | ✓ | ✓ | ✓ | - | 92 | 76.6 | 89.2 | 0.84 | 38.02 |
| ✓ | ✓ | ✓ | ✓ | ✓ | 91.5 | 77.6 | 89.4 | 0.84 | 60.98 |

resource-constrained devices. To mitigate this, we optimized the model design by combining Swin Transformer with efficient Depthwise Convolution to reduce computation while preserving spatial information. We also introduced the SPPCSPC module to enhance multi-scale feature fusion in a parameter-efficient manner, significantly improving detection speed.

Despite the high accuracy achieved, further improvements can be made. The model struggles with highly occluded buds and still misclassifies some leaves as buds. Additional contextual and shape information may help overcome this. Integrating multimodal data sources like infrared or depth images may also enhance robustness. On the optimization front, techniques like neural architecture search could help find designs even better suited for this task. Deployment-specific optimizations like quantization-aware training can reduce the computational requirements.

Recent related works have made progress in small object detection. Zhang, et al. [28] detected crop buds by adding a micro-prediction head and attention modules. Shuai, et al. [30] explored multimodal fusion for shoot detection. These inspire ideas like employing normalized losses for increased robustness and leveraging. Carefully reviewing such works can better contextualize our model's contributions and limitations.

In terms of practical deployment, the choice of hardware is key. Compact embedded devices would enable onboard detection on UAVs for automated monitoring. Edge servers can provide low-latency inference by locating computation closer to the sensing devices. The algorithm could also be integrated into larger agricultural intelligence systems, combining environmental data for precision management. Further developing supporting decision-making and control software can transform this technology into solutions that increase productivity.

In conclusion, this study makes notable progress in the challenging problem of tiny object detection for tea buds and plants in unconstrained natural environments. Our model delivers

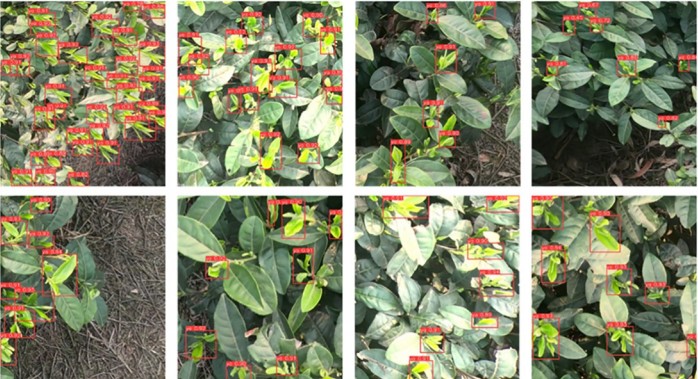

**Fig 13. Detecting recognition effect.**

state-of-the-art accuracy while being cognizant of efficiency constraints. Nonetheless, we identify multiple promising directions for improvement through architectural enhancements, supplementary data sources, and practical deployment optimizations. The proposed approach and analyses contribute valuable insights toward enabling automated vision systems for agriculture.

## Conclusion

This study addresses the problem of detecting tiny tea buds for agricultural monitoring, which poses multiple challenges like small target sizes, dense object distribution, complex backgrounds, and morphological similarities with other components. Our proposed STF-YOLO model delivers state-of-the-art accuracy by effectively incorporating transformer-based attention for enhanced feature representation while using efficient convolutional designs to maintain reasonable detection speeds. Specifically, STF-YOLO achieves a precision of 91.5%, recall of 77.6%, mAP@0.5 of 89.4%, and mAP@0.5:0.95 of 71% on our tea bud dataset. This signifies substantial improvements over prior YOLO variants as well as other detection models.

However, limitations exist in terms of model size and complexity. The incorporation of additional components results in STF-YOLO not being lightweight enough for highly resource-constrained environments. Handling highly occluded cases and distinguishing buds from similar leaves also needs improvement.

Future work should focus on further compressing the model design as well as incorporating additional shape and context information and exploring supplementary data sources. On the system's front, optimizations targeted at embedded deployment can help realize practical UAV and edge computing solutions. Nonetheless, this research makes notable progress in tiny object detection, with the presented approach, analyses, and directions laying the groundwork to enable automated vision for agriculture.

## Acknowledgments

The author would like to express thanks to anonymous reviewers for all careful review of the paper and kind suggestions made to improve overall quality of the manuscript.

## Author Contributions

**Conceptualization:** Meiling Shi.

**Data curation:** Meiling Shi.

**Formal analysis:** Meiling Shi.

**Funding acquisition:** Kailiang Huang.

**Investigation:** Meiling Shi.

**Methodology:** Meiling Shi.

**Project administration:** Meiling Shi, Kailiang Huang.

**Resources:** Meiling Shi, Tianhao Wu, Wenjing Zhang, Kailiang Huang.

**Software:** Meiling Shi.

**Supervision:** Meiling Shi.

**Validation:** Meiling Shi, Tianhao Wu, Wenjing Zhang.

**Visualization:** Meiling Shi, Tianhao Wu.

**Writing – original draft:** Meiling Shi, Dongling Zheng, Tianhao Wu, Wenjing Zhang, Ruijie Fu.

**Writing – review & editing:** Meiling Shi, Dongling Zheng.

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
