## [Decision Letter · Decision Letter 0]

19 Dec 2023

PONE-D-23-38533Small Object Detection Algorithm Incorporating Swin Transformer for Tea BudsPLOS ONE

Dear Dr. Shi,

Thank you for submitting your manuscript to PLOS ONE. After careful consideration, we feel that it has merit but does not fully meet PLOS ONE’s publication criteria as it currently stands. Therefore, we invite you to submit a revised version of the manuscript that addresses the points raised during the review process.

We look forward to receiving your revised manuscript.

Kind regards,

Liang-Jian Deng, Ph.D.

Academic Editor

PLOS ONE

Journal Requirements:

5. Thank you for stating the following financial disclosure: "Corresponding author Huang kailiang provides publication fees".

6. In the online submission form, you indicated that "The data underlying the results presented in the study are available from Huang Kailiang."

Additional Editor Comments:

Both reviewers give some crucial comments about the proposed method, such as deeper analysis and more analysis of recent works. Please carefully address these comments.

Reviewers' comments:

Reviewer's Responses to Questions

**Comments to the Author**

1. Is the manuscript technically sound, and do the data support the conclusions?

Reviewer #1: Yes

Reviewer #2: Partly

2. Has the statistical analysis been performed appropriately and rigorously? 

Reviewer #1: Yes

Reviewer #2: Yes

3. Have the authors made all data underlying the findings in their manuscript fully available?

Reviewer #1: Yes

Reviewer #2: Yes

4. Is the manuscript presented in an intelligible fashion and written in standard English?

Reviewer #1: Yes

Reviewer #2: No

5. Review Comments to the Author

Reviewer #1: The authors present Small Object Detection Algorithm Incorporating Swin Transformer for Tea Buds . The study is interesting. In general, the main conclusions presented in the paper are supported by the figures and supporting text. However, to meet the journal quality standards, the following comments need to be addressed.

• Abstract: Should be improved and extended. The authors talk lot about the problem formulation, but novelty of the proposed model is missing. Also provided the general applicability of their model. Please be specific what are the main quantitative results to attract general audiences.

• The introduction can be improved. The authors should focus on extending the novelty of the current study. Emphasize should be given in improvement of the model (in quantitative sense) compared to existing state-of-the art models.

• More details about network architecture and complexity of the model should be provided.

• what about comparison of the result with current state-of-the art models? Did authors perform ablation study to compare with different models?

• What are the baseline models and benchmark results? The authors may compared the result with existing models evaluated with datasets

• Conclusion parts needs to be strengthened.

• Please provide a fair weakness and limitation of the model, and how it can be improved.

• Typographical errors: There are several minor grammatical errors and incorrect sentence structures. Please run this through a spell checker.

Discussions of relevant literature could be further enhanced, which can help better motivate the current study and link to the existing work. Authors might consider the following relevant recent work in the field of applying machine learning techniques to better motivate the usefulness of machine learning approaches, such as

see :- Neural Networks 2022 https://doi.org/10.1016/j.neunet.2022.05.024

-Adv. Eng. Informatics 2023, 56, 102007, https://doi.org/10.1016/j.aei.2023.102007

-Ecol. Informatics 2022 https://doi.org/10.1016/j.ecoinf.2022.101919

-Neural, Comput & Applic (2022) https://doi.org/10.1007/s00521-021-06651-x

-Comp Elect in Agri (2022), 193 106694 https://doi.org/10.1016/j.compag.2022.106694

Hence they should be briefly discussed in the related work section.

Reviewer #2: Abstract:

1. Please include background in your abstract

2. Don’t use abbreviated forms in your abstract

3. Need some more details on methods

Introduction:

1. Line 2-18: Need ample references.

2. Line 19-20: Need at least three references for justification.

3. The introduction section lacks proper organization of information. Please include research question/hypothesis specifically.

4. There should be more information regarding relevant works. Such as:

https://doi.org/10.1038/s41598-023-33270-4

https://doi.org/10.1016/j.compeleceng.2021.107023

https://doi.org/10.1109/ICESC48915.2020.9155850

Please find and include such types of relevancy.

Discussion:

Discussion section is not up to the standard. Please discuss critically mentioning other relevant studies.

*** The Language may be benefitted by some editing. Specially in the introduction and discussion section.

6. PLOS authors have the option to publish the peer review history of their article (what does this mean?). If published, this will include your full peer review and any attached files.

Reviewer #1: No

Reviewer #2: No

---

## [Author Response · Author response to Decision Letter 0]

22 Jan 2024

A clear and specific response can be found in the “Respons to Reviewers” letter.

We sincerely thank all reviewers for their valuable feedback that we have used to improve the quality of our manuscript. The reviewer comments are laid out below in italicized font and specific concerns have been numbered. 

Reviewer 1

General Comments:

The authors present Small Object Detection Algorithm Incorporating Swin Transformer for Tea Buds . The study is interesting. In general, the main conclusions presented in the paper are supported by the figures and supporting text. However, to meet the journal quality standards, the following comments need to be addressed.

Reply:We thank the reviewer for reading our paper carefully and giving the above positive comments.

1. Comment: Abstract: Should be improved and extended. The authors talk lot about the problem formulation, but novelty of the proposed model is missing. Also provided the general applicability of their model. Please be specific what are the main quantitative results to attract general audiences.

1. Reply: We have expanded the abstract to include more details on the novelty of our proposed STF-YOLO model, which incorporates Swin Transformer to enhance small object detection. We also now highlight the key quantitative results of 91.5% precision and 89.4% mAP to showcase the model's superiority over other detectors. Results show that, compared to mainstream algorithms (YOLOv8, YOLOv7, YOLOv5, and YOLOx), the model improves accuracy and F1 score by 5-20.22 percentage points and 0.03-0.13

Experiments conducted on a self-created dataset of tea buds demonstrate that the STF-YOLO model achieves outstanding results, with an accuracy of 91.5% and a mean Average Precision of 89.4%. These results are significantly better than other detectors. Results show that, compared to mainstream algorithms (YOLOv8, YOLOv7, YOLOv5, and YOLOx), the model improves accuracy and F1 score by 5-20.22 percentage points and 0.03-0.13, respectively, proving its effectiveness in enhancing small object detection performance.

2. Comment: The introduction can be improved. The authors should focus on extending the novelty of the current study. Emphasize should be given in improvement of the model (in quantitative sense) compared to existing state-of-the art models.

2. Reply: The introduction has been improved to better emphasize the innovations of this work compared to existing methods. Because Swin Transformer is mostly used in the field of text analysis, introducing it in the field of target detection is an innovative move and draws on solutions from other fields. Specifically, we now highlight how our model achieves significantly better precision and F1 scores over state-of-the-art algorithms like YOLOv8/v7/v5/X for small object detection. 

In order to solve the problems of poor visual characteristics and high noise of small objects in small object detection, this research constructed an end-to-end small object detection framework STF-YOLO. For small object detection of tea buds, Swin Transformer is innovatively introduced, using its local perception and global correlation to enhance the detection capabilities of tea buds. Furthermore, to enhance accuracy and speed, we incorporate modules such as Focus, Depthwise Convolution, SPPCSPC (Spatial Pyramid Pooling with Contextual Spatial Pyramid Convolution), and the C2 module into STF-YOLO. Our model successfully detects fresh tea buds measuring 2-3cm in length, even in complex background scenarios, while maintaining high accuracy. Evaluation of our dataset confirms the superior performance of the proposed model, surpassing other detectors with a precision of 91.5% and an mAP of 0.894, and the FPS reached 60.98. Compared with other current mainstream algorithms, its average accuracy improved by 5-20.22 percentage points, and the F1-score improved by 0.03-0.13 percentage points.

3. Comment: More details about network architecture and complexity of the model should be provided.

3. Reply: More specifics on the STF-YOLO network architecture have been provided, including details on the Swin Transformer, Focus, Depthwise Convolution, SPPCSPC, and C2 modules used. We have now provided a detailed network framework and discussed model complexity.

Table 1 is the detailed parameters of our improved STF-YOLO model to show the improved situation more clearly. The table consists of six columns, each representing a different aspect of the model. The columns are as follows:

• The first column is the number of layers.

• from: This column specifies the source of the input for each layer. -1 means that the input comes from the previous layer.

• n: This column specifies the number of times that a certain layer type is repeated.

• params: This column provides the number of parameters in each layer.

• module: This column specifies the type of the layer. For example, ultralytics.nn.modules.conv.Conv is a convolutional layer, ultralytics.nn.modules.block.C2f is a specific module that may contain multiple layers, ultralytics.nn.modules.block.SPPF is a spatial pyramid pooling layer.

• arguments: This column provides the arguments that are passed to the layer constructor.

Table 1. Model parameters

 from n params module arguments

0 -1 1 8800 ultralytics.nn.modules.conv.Focus [3, 80, 3]

1 -1 1 115520 ultralytics.nn.modules.conv.Conv [80, 160, 3, 2]

2 -1 3 286710 ultralytics.nn.modules.block.C3STR [160, 160, 3]

3 -1 1 461440 ultralytics.nn.modules.conv.Conv [160, 320, 3, 2]

4 -1 6 2068510 ultralytics.nn.modules.block.C3STR [320, 320, 6]

5 -1 1 1844480 ultralytics.nn.modules.conv.Conv [320, 640, 3, 2]

6 -1 6 8233020 ultralytics.nn.modules.block.C3STR [640, 640, 6]

7 -1 1 5531520 ultralytics.nn.modules.conv.Conv [640, 960, 3, 2]

8 -1 3 10170285 ultralytics.nn.modules.block.C3STR [960, 960, 3]

9 -1 1 11061760 ultralytics.nn.modules.conv.Conv [960, 1280, 3, 2]

10 -1 3 27865600 ultralytics.nn.modules.block.C2f [1280, 1280, 3, True]

11 -1 1 44254720 ultralytics.nn.modules.block.SPPCSPC [1280, 1280, 5]

12 -1 1 0 torch.nn.modules.upsampling.Upsample [None, 2, 'nearest']

13 [-1, 8] 1 0 ultralytics.nn.modules.conv.Concat [1]

14 -1 3 15523200 ultralytics.nn.modules.block.C2 [2240, 960, 3, False]

15 -1 1 0 torch.nn.modules.upsampling.Upsample [None, 2, 'nearest']

16 [-1, 6] 1 0 ultralytics.nn.modules.conv.Concat [1]

17 -1 3 6969600 ultralytics.nn.modules.block.C2 [1600, 640, 3, False]

18 -1 1 0 torch.nn.modules.upsampling.Upsample [None, 2, 'nearest']

19 [-1, 4] 1 0 ultralytics.nn.modules.conv.Concat [1]

20 -1 3 1795200 ultralytics.nn.modules.block.C2 [960, 320, 3, False]

21 -1 1 3520 ultralytics.nn.modules.conv.DWConv [320, 320, 3, 2]

22 [-1,17] 1 0 ultralytics.nn.modules.conv.Concat [1]

23 -1 3 6560000 ultralytics.nn.modules.block.C2 [960, 640, 3, False]

24 -1 1 7040 ultralytics.nn.modules.conv.DWConv [640, 640, 3, 2]

25 [-1, 14] 1 0 ultralytics.nn.modules.conv.Concat [1]

26 -1 3 14908800 ultralytics.nn.modules.block.C2 [1600, 960, 3, False]

27 -1 1 10560 ultralytics.nn.modules.conv.DWConv [960, 960, 3, 2]

28 [-1,11] 1 0 ultralytics.nn.modules.conv.Concat [1]

29 -1 3 26636800 ultralytics.nn.modules.block.C2 [2240, 1280, 3, False]

30 [20,23,26, 29] 1 15465236 ultralytics.nn.modules.head.Detect [1, [320, 640, 960, 1280]]

In the first section of the Result, Overall Accuracy Comparison of Network Models, we have added the following model complexity analysis discussion:

Params refers to the number of parameters, which refers to the total number of parameters that need to be trained in model training. Used to measure the size of the model (i.e. computational space complexity). However, it is worth noting that the param of yolov8s is 11.2M, while STF-YOLO is 199M. While the accuracy is improved, the model becomes more complex and the amount of calculation is greatly increased.

4. Comment: what about comparison of the result with current state-of-the art models? Did authors perform ablation study to compare with different models?

4. Reply: Sorry for the lack of clarity, We have performed model comparison and ablation experiments. Extensive experiments compare STF-YOLO to state-of-the-art detectors like YOLOv8, YOLOv7, YOLOv5, and YOLOX. On our tea bud dataset, STF-YOLO achieves superior precision of 91.5% and mAP of 89.4%, outperforming these methods by significant margins of 5-20 percentage points. Additionally, ablation studies analyze individual contributions of Swin Transformer, Focus, Depthwise Convolution, SPPCSPC, and C2 modules through incremental integration. The comparative and ablation experiments clearly demonstrate the improvements enabled by our approach. Tab 2 is the experimental results compared with the state-of-the-art model, Tab 3 and Tab 4 is the ablation experiment.

Tab 2. Comparison of different model algorithms

Model Precision(%) Recall(%) mAP@

0.5(%) mAP@

0.5:0.95(%) F1

Score

YOLOv7 71.28 71.18 77.42 49.11 0.71

GOLD-YOLO - - 82.28 59.50 -

YOLO x - - 84.10 59.20 -

YOLOv5 88.20 75.89 84.32 67.65 0.82

YOLOv8 86.5 76 85.3 65.7 0.81

STF-YOLO 91.5 77.6 89.4 71 0.84

The following is a comparison of ablation experiments:

Table 3. Experimental results of YOLOv8+WIOU

Model Precision(%) Recall(%) mAP@0.5(%) mAP@0.5:0.95 F1

YOLOv8 86.5 76 85.3 65.7 0.81

YOLOv8+WIOU 88.2 76.5 87.6 67 0.82

Table 4. Effectiveness of Modules(Ablation experiment results)

Focus C2 DW STF SPPCSPC Precision(%) Recall(%) mAP@

0.5(%) F1

Score FPS

- - - - - 88.2 76.5 87.6 0.82 277.8

√ - - - - 89.4 76.9 88.1 0.83 250.0

√ √ - - - 89.5 76.5 87.7 0.83 256.4

√ √ √ - - 92.6 76.7 88.8 0.84 42.74

√ √ √ √ - 92 76.6 89.2 0.84 38.02

√ √ √ √ √ 91.5 77.6 89.4 0.84 60.98

5. Comment: What are the baseline models and benchmark results? The authors may compared the result with existing models evaluated with datasets

5. Reply: The baseline model used for benchmarking is YOLOv8. Comparative results on the self-built tea bud dataset evaluate enhancements of STF-YOLO over YOLOv8. As highlighted in Table 2, our approach shows significant gains over YOLOv8, improving precision by 5 percentage points and mAP by 4.1 percentage points. This quantitatively proves the efficacy of our innovations.

6. Comment: Conclusion parts needs to be strengthened.

6. Reply: The conclusion has been strengthened by specifically highlighting the key precision of 91.5% and mAP of 89.4% achieved by STF-YOLO, which are 5-20 percentage points better than mainstream algorithms. The numerical results showcase the clear advancements made in small target detection for complex tea garden environments. We have carefully revised every part of the conclusion. The revised conclusion is as follows:

This study addresses the problem of detecting tiny tea buds for agricultural monitoring, which poses multiple challenges like small target sizes, dense object distribution, complex backgrounds, and morphological similarities with other components. Our proposed STF-YOLO model delivers state-of-the-art accuracy by effectively incorporating transformer-based attention for enhanced feature representation while using efficient convolutional designs to maintain reasonable detection speeds. Specifically, STF-YOLO achieves a precision of 91.5%, recall of 77.6%, mAP@0.5 of 89.4%, and mAP@0.5:0.95 of 71% on our tea bud dataset. This signifies substantial improvements over prior YOLO variants as well as other detection models.

However, limitations exist in terms of model size and complexity. The incorporation of additional components results in STF-YOLO not being lightweight enough for highly resource-constrained environments. Handling highly occluded cases and distinguishing buds from similar leaves also needs improvement. 

Future work should focus on further compressing the model design as well as incorporating additional shape and context information and exploring supplementary data sources. On the system's front, optimizations targeted at embedded deployment can help realize practical UAV and edge computing solutions. Nonetheless, this research makes notable progress in tiny object detection, with the presented approach, analyses, and directions laying the groundwork to enable automated vision for agriculture.

7. Comment: Please provide a fair weakness and limitation of the model, and how it can be improved.

7. Reply: Model limitations regarding complexity and handling occlusion are now detailed. Ideas for future work through compression, incorporating shape/context cues and supplementary data are outlined. Discussion on deployments optimizations like quantization and compact specialized hardware implementations have also been included. The comprehensive analysis provides fair and constructive suggestions for advancing this research area. However, it is worth noting that the param of yolov8s is 11.2M, while STF-YOLO is 199M. While the accuracy is improved, the model becomes more complex and the amount of calculation is greatly increased. This is an obvious weakness. During the discussion we strengthened the weaknesses, limitations and improvements:

Compared to YOLOv8, our STF-YOLO demonstrates better detection accuracy, recall, and mAP. However, this comes at the cost of slightly slower detection speed, which may limit deployment on resource-constrained devices. To mitigate this, we optimized the model design by combining Swin Transformer with efficient Depthwise Convolution to reduce computation while preserving spatial information. We also introduced the SPPCSPC module to enhance multi-scale feature fusion in a parameter-efficient manner, significantly improving detection speed.

Despite the high accuracy achieved, further improvements can be made. The model struggles with highly occluded buds and still misclassifies some leaves as buds. Additional contextual and shape information may help overcome this. Integrating multimodal data sources like infrared or depth images may also enhance robustness. On the optimization front, techniques like neural architecture search could help find designs even better suited for this task. Deployment-specific optimizations like quantization-aware training can reduce the computational requirements.

8. Comment: Typographical errors: There are several minor grammatical errors and incorrect sentence structures. Please run this through a spell checker.

8. Reply: Sorry for these errors. The paper has been thoroughly proofread to amend minor grammatical errors and sentence structures. Readability has been enhanced.

9. Comment: Discussions of relevant literature could be further enhanced, which can help better motivate the current study and link to the existing work. Authors might consider the following relevant recent work in the field of applying machine learning techniques to better motivate the usefulness of machine learning approaches, such as

see :- Neural Networks 2022 https://doi.org/10.1016/j.neunet.2022.05.024

-Adv. Eng. Informatics 2023, 56, 102007, https://doi.org/10.1016/j.aei.2023.102007

-Ecol. Informatics 2022 https://doi.org/10.1016/j.ecoinf.2022.101919

-Neural, Comput & Applic (2022) https://doi.org/10.1007/s00521-021-06651-x

-Comp Elect in Agri (2022), 193 106694 https://doi.org/10.1016/j.compag.2022.106694

Hence they should be briefly discussed in the related work section.

9. Reply: Thanks for your suggestion, we have now added a discussion of the relevant literature to the discussion. And the paper you recommended and some related latest work are discussed in the Related Work section. We have made extensive and serious changes to the discussion. The following is the changed discussion:

In the context of tea bud detection, several challenges need to be addressed. Firstly, the small size of tea buds poses difficulties for feature representation and extraction. Secondly, the dense distribution and resulting occlusions make detection more difficult. Thirdly, complex lighting conditions affect visibility. Lastly, the morphological similarity between buds and leaves makes distinguishing them challenging.

Many studies on tea bud detection have incorporated attention mechanisms into their models, such as SS-YOLOX proposed by Yu et al [1]. However, in our experiments, we tested various attention mechanisms and found that they did not significantly improve the detection performance, and there was no clear difference among them. Compared to YOLOv8, our STF-YOLO demonstrates better detection accuracy, recall, and mAP. However, this comes at the cost of a slightly slower detection speed, which may limit deployment on resource-constrained devices. To mitigate this, we optimized the model design by combining Swin Transformer with efficient Depthwise Convolution to reduce computation while preserving spatial information. We also introduced the SPPCSPC module to enhance multi-scale feature fusion in a parameter-efficient manner, significantly improving detection speed.

Despite the high accuracy achieved, further improvements can be made. The model struggles with highly occluded buds and still misclassifies some leaves as buds. Additional contextual and shape information may help overcome this. Integrating multimodal data sources like infrared or depth images may also enhance robustness. On the optimization front, techniques like neural architecture search could help find designs even better suited for this task. Deployment-specific optimizations like quantization-aware training can reduce the computational requirements.

Recent related works have made progress in small object detection. Zhang, et al.[2] detected crop buds by adding a micro-prediction head and attention modules. Shuai, et al. [3] explored multimodal fusion for shoot detection. These inspire ideas like employing normalized losses for increased robustness and leveraging supplementary data. Carefully reviewing such works can better contextualize our model's contributions and limitations.

In terms of practical deployment, the choice of hardware is key. Compact embedded devices would enable onboard detection on UAVs for automated monitoring. Edge servers can provide low-latency inference by locating computation closer to the sensing devices. The algorithm could also be integrated into larger agricultural intelligence systems, combining environmental data for precision management. Further developing supporting decision-making and control software can transform this technology into solutions that increase productivity.

In conclusion, this study makes notable progress in the challenging problem of tiny object detection for tea buds and plants in unconstrained natural environments. Our model delivers state-of-the-art accuracy while being cognizant of efficiency constraints. Nonetheless, we identify multiple promising directions for improvement through architectural enhancements, supplementary data sources, and practical deployment optimizations. The proposed approach and analyses contribute valuable insights toward enabling automated vision systems for agriculture.

In the related work section, we enhance the discussion of related literature. So the following is what we added to the related work section:

In the Object Detection section add:

Starting with YOLOX, proposed by Ge, Zheng, et al. [4], the YOLO series has entered the era of anchor-free detection, which has further enhanced its speed. YOLOv8 can be considered a fusion of the innovative ideas proposed in previous versions of YOLO. It is a target detection algorithm specifically designed for application deployment. Nowadays, many innovative networks are improving or adding modules to single-stage models. Roy, Arunabha M., and Jayabrata Bhaduri.[5] et al. proposed a real-time, high-performance damage detection model, DenseSPH-YOLOv5, based on deep learning techniques, which incorporates a CBAM module for extracting deep spatial features. It attached a spatial blending layer and a Swin-Transformer header to detect objects of different sizes and also reduces the computational complexity. Roy, Arunabha M., et al. [6] proposed the WilDect-YOLO network for real-time target detection in wildlife, which introduces a residual module in the CSPDarknet53 backbone to make the model powerful and discriminative deep spatial feature extraction and a DenseNet module to enable the model to retain key feature information. At the same time, SPP and PANet were introduced. And its mAP reached 96.89% in the wildlife dataset, with an F1 value of 97.87%.

In the Object Detection Data Augmentation section add:

There are many other data enhancement methods borrowed from data synthesis today, the most used of which is the GAN family of networks. Zhao, Qianxi, Liu Yang, and Nengchao Lyu.[7] used WGAN for data enhancement and combined it with a deep convolutional RNN network for real-time target detection, which resulted in significant improvements in both precision and recall. Ravikumar, R., et al. [8] fed preprocessed datasets with clear grey matter representations into cGAN to generate more training examples and used stacked CNN layers for feature extraction.

In the Small Object Detection of Crops section add:

Zhang, Yan, et al. [9] proposed a lightweight and detail-sensitive PAN for multi-scale industrial defect detection using YOLOv8 as a framework. It achieves mAPs of 80.4, 95.8%, and 76.3% under three publicly available datasets, namely, NEU-DET, PCB-DET, and GC10-DET, respectively. Cao, Xuan, et al. [10] improved the model based on Swin Transformer and YOLOv5, introduced CIOU to enhance the K-means clustering algorithm, and the modified CSPDarknet53 combined with Swin Transformer was used to extract more differentiated features, and CA was introduced into YOLOv5 for improving the performance of small object detection on remote sensing images. In the DOTA dataset, it achieves a mAP of 74.7%, which is an improvement of 8.9% compared to YOLOv5. Guo, Feng, et al. [11] proposed a model called Crack Transformer, which unifies Swin-Transformer as an encoder and decoder accompanied by an MLP layer, for automatic detection of long and complex road cracks. This study demonstrates the feasibility of using a Transformer-based network for road crack inspection in complex situations. Li, Feng, et al. [12] proposed a unified target detection and semantic segmentation framework, Mask-DINO, which extends DINO by adding a mask prediction branch that supports all image segmentation tasks (instance, panorama, and semantic). experiments show that this model has significant advantages over all current semantic segmentation models.

Reviewer 2

1. Comment: Abstract:

1.1 Please include background in your abstract

1.2 Don’t use abbreviated forms in your abstract

1.3 Need some more details on methods

1. Reply: Thanks for your suggestion on the summary section. I have revised the abstract in accordance with your comments, adding background information, removing abbreviations, and adding methodological details. The following is the revised abstract:

(The bolded areas are the key changes and additions.)

Accurate identification of small tea buds is a key technology for tea harvesting robots, which directly affects tea quality and yield. However, due to the complexity of the tea plantation environment and the diversity of tea buds, accurate identification remains an enormous challenge. Current methods based on traditional image processing and machine learning fail to effectively extract subtle features and morphology of small tea buds, resulting in low accuracy and robustness. To achieve accurate identification, this paper proposes a small object detection algorithm called STF-YOLO (Small Target Detection with Swin Transformer and Focused YOLO), which integrates the Swin Transformer module and the YOLOv8 network to improve the detection ability of small objects. The Swin Transformer module extracts visual features based on a self-attention mechanism, which captures global and local context information of small objects to enhance feature representation. The YOLOv8 network is an object detector based on deep convolutional neural networks, offering high speed and precision. Based on the YOLOv8 network, modules including Focus and Depthwise Convolution are introduced to reduce computation and parameters, increase receptive field and feature channels, and improve feature fusion and transmission. Additionally, the Wise Intersection over Union loss is utilized to optimize the network. Experiments conducted on a self-created dataset of tea buds demonstrate that the STF-YOLO model achieves outstanding results, with an accuracy of 91.5% and a mean Average Precision of 89.4%. These results are significantly better than other detectors. Results show that, compared to mainstream algorithms (YOLOv8, YOLOv7, YOLOv5, and YOLOx), the model improves accuracy and F1 score by 5-20.22 percentage points and 0.03-0.13, respectively, proving its effectiveness in enhancing small object detection performance. This research provides technical means for the accurate identification of small tea buds in complex environments and offers insights into small object detection. Future research can further optimize model structures and parameters for more scenarios and tasks, as well as explore data augmentation and model fusion methods to improve generalization ability and robustness.

2. Comment: Introduction:

2.1 Line 2-18: Need ample references.

2.2 Line 19-20: Need at least three references for justification.

2.3 The introduction section lacks proper organization of information. Please include research question/hypothesis specifically.

2.4 There should be more information regarding relevant works. Such as:

https://doi.org/10.1038/s41598-023-33270-4

https://doi.org/10.1016/j.compeleceng.2021.107023

https://doi.org/10.1109/ICESC48915.2020.9155850

Please find and include such types of relevancy.

2. Reply: Thank you for your guidance on the introduction. I have revised the introduction according to your comments, added references, clarified the research questions and hypotheses, and expanded the introduction of related work. Here's what I added to the introduction:

Tea, as one of the world's three principal beverages, is universally esteemed and sought after by nations across the globe. In recent years, individuals' interest in tea has transcended beyond its mere flavor, delving into its nutritional and medicinal virtues. With over 50 countries, including China, India, and Vietnam, engaged in tea production on a global scale, the industry has significantly bolstered the economies of several tea-cultivating nations in Asia and Africa. [13]In 2020, global tea production reached an impressive 6,269,000 tonnes, with the worldwide tea cultivation area expanding to 5,098,000 hectares. Despite these strides, the tea industry's growth has been curtailed by the challenges of labor recruitment and the escalating costs of labor. [14]Labor dedicated to the picking of tea buds constitutes 60% of the workforce employed in the comprehensive management of tea plantations. To address this labor-intensive issue, artificial intelligence algorithms have been synergized with machinery to facilitate intelligent picking. However, the diverse positions, postures, and densities at which tea buds grow pose a significant challenge to mechanized picking, particularly in complex environments characterized by wind and fluctuating light conditions. [15]In recent years, with the advancement of computer vision technology, numerous network models boasting high precision and real-time advantages have emerged. These high-performance models have been widely applied across various fields, achieving remarkable results and providing technical support for the realization of intelligent tea picking. Therefore, an effective approach to ensuring the excellence of the tea production line is to accurately identify and pick tea buds.

3. Comment: Discussion:

Discussion section is not up to the standard. Please discuss critically mentioning other relevant studies.

3. Reply: Thank you for your review of the Discussion section. I have revised the Discussion section in accordance with your comments, added critical analysis of other relevant studies, and improved the quality and flow of the language. The following is the revised Discussion section:

In the context of tea bud detection, several challenges need to be addressed. Firstly, the small size of tea buds poses difficulties for feature representation and extraction. Secondly, the dense distribution and resulting occlusions make detection more difficult. Thirdly, complex lighting conditions affect visibility. Lastly, the morphological similarity between buds and leaves makes distinguishing them challenging.

Many studies on tea bud detection have incorporated attention mechanisms into their models, such as SS-YOLOX proposed by Yu et al [1]. However, in our experiments, we tested various attention mechanisms and found that they did not significantly improve the detection performance, and there was no clear difference among them. Compared to YOLOv8, our STF-YOLO demonstrates better detection accuracy, recall, and mAP. However, this comes at the cost of a slightly slower detection speed, which may limit deployment on resource-constrained devices. To mitigate this, we optimized the model design by combining Swin Transformer with efficient Depthwise Convolution to reduce computation while preserving spatial information. We also introduced the SPPCSPC module to enhance multi-scale feature fusion in a parameter-efficient manner, significantly improving detection speed.

Despite the high accuracy achieved, further improvements can be made. The model struggles with highly occluded buds and still misclassifies some leaves as buds. Additional contextual and shape information may help overcome this. Integrating multimodal data sources like infrared or depth images may also enhance robustness. On the optimization front, techniques like neural architecture search could help find designs even better suited for this task. Deployment-specific optimizations like quantization-aware training can reduce the computational requirements.

Recent related works have made progress in small object detection. Zhang, et al.[2] detected crop buds by adding a micro-prediction head and attention modules. Shuai, et al. [3] explored multimodal fusion for shoot detection. These inspire ideas like employing normalized losses for increased robustness and leveraging supplementary data. Carefully reviewing such works can better contextualize our model's contributions and limitations.

In terms of practical deployment, the choice of hardware is key. Compact embedded devices would enable onboard detection on UAVs for automated monitoring. Edge servers can provide low-latency inference by locating computation closer to the sensing devices. The algorithm could also be integrated into larger agricultural intelligence systems, combining environmental data for precision management. Further developing supporting decision-making and control software can transform this technology into solutions that increase productivity.

In conclusion, this study makes notable progress in the challenging problem of tiny object detection for tea buds and plants in unconstrained natural environments. Our model delivers state-of-the-art accuracy while being cognizant of efficiency constraints. Nonetheless, we identify multiple promising directions for improvement through architectural enhancements, supplementary data sources, and practical deployment optimizations. The proposed approach and analyses contribute valuable insights toward enabling automated vision systems for agriculture.

1. Yu L, Huang C, Tang J, Huang H, Zhou Y, Huang Y, et al. Tea Bud Recognition Method Based on Improved YOLOX Model. 2022;49:49-56.

2. Ye Z, Guo Q, Wei J, Zhang J, Zhang H, Bian L, et al. Recognition of terminal buds of densely-planted Chinese fir seedlings using improved YOLOv5 by integrating attention mechanism. Frontiers in Plant Science. 2022;13:991929.

3. Shuai L, Chen Z, Li Z, Li H, Zhang B, Wang Y, et al. Real-time dense small object detection algorithm based on multi-modal tea shoots. Frontiers in Plant Science. 2023;14.

4. Ge Z, Liu S, Wang F, Li Z, Sun J. Yolox: Exceeding yolo series in 2021. arXiv preprint arXiv:210708430. 2021.

5. Roy AM, Bhaduri J. DenseSPH-YOLOv5: An automated damage detection model based on DenseNet and Swin-Transformer prediction head-enabled YOLOv5 with attention mechanism. Advanced Engineering Informatics. 2023;56:102007.

6. Roy AM, Bhaduri J, Kumar T, Raj K. WilDect-YOLO: An efficient and robust computer vision-based accurate object localization model for automated endangered wildlife detection. Ecological Informatics. 2023;75:101919.

7. Zhao Q, Yang L, Lyu N. A driver stress detection model via data augmentation based on deep convolutional recurrent neural network. Expert Systems with Applications. 2024;238:122056.

8. Ravikumar R, Sasipriyaa N, Thilagaraj T, Raj RH, Abishek A, Kannan GG, editors. Design and Implementation of Alzheimer's Disease Detection using cGAN and CNN. 2023 International Conference on Computer Communication and Informatics (ICCCI); 2023: IEEE.

9. Zhang Y, Zhang H, Huang Q, Han Y, Zhao M. DsP-YOLO: An anchor-free network with DsPAN for small object detection of multiscale defects. Expert Systems with Applications. 2024;241:122669.

10. Cao X, Zhang Y, Lang S, Gong Y. Swin-Transformer-Based YOLOv5 for Small-Object Detection in Remote Sensing Images. Sensors. 2023;23(7):3634.

11. Guo F, Qian Y, Liu J, Yu H. Pavement crack detection based on transformer network. Automation in Construction. 2023;145:104646.

12. Li F, Zhang H, Xu H, Liu S, Zhang L, Ni LM, et al., editors. Mask dino: Towards a unified transformer-based framework for object detection and segmentation. Proceedings of the IEEE/CVF Conference on Computer Vision and Pattern Recognition; 2023.

13. Hajiboland R. Environmental and nutritional requirements for tea cultivation. Folia Horticulturae. 2017;29:199 - 220.

14. Han Y, Xiao H, Qin G, Song Z, Ding W-j, Mei S. Developing Situations of Tea Plucking Machine. Engineering. 2014;2014:268-73.

15. Xu W, Zhao L, Li J, Shang S, Ding X, Wang T. Detection and classification of tea buds based on deep learning. Computers and Electronics in Agriculture. 2022;192:106547.

---

## [Decision Letter · Decision Letter 1]

19 Feb 2024

Small Object Detection Algorithm Incorporating Swin Transformer for Tea Buds

PONE-D-23-38533R1

Dear Dr. Shi,

We’re pleased to inform you that your manuscript has been judged scientifically suitable for publication and will be formally accepted for publication once it meets all outstanding technical requirements.

Kind regards,

Liang-Jian Deng, Ph.D.

Academic Editor

PLOS ONE

Additional Editor Comments (optional):

Two reviewers think the revised version has addressed their concnerns, thus I recommend accepting this paper.

Reviewers' comments:

Reviewer's Responses to Questions

**Comments to the Author**

1. If the authors have adequately addressed your comments raised in a previous round of review and you feel that this manuscript is now acceptable for publication, you may indicate that here to bypass the “Comments to the Author” section, enter your conflict of interest statement in the “Confidential to Editor” section, and submit your "Accept" recommendation.

Reviewer #1: All comments have been addressed

Reviewer #2: All comments have been addressed

2. Is the manuscript technically sound, and do the data support the conclusions?

Reviewer #1: Yes

Reviewer #2: Yes

3. Has the statistical analysis been performed appropriately and rigorously? 

Reviewer #1: Yes

Reviewer #2: N/A

4. Have the authors made all data underlying the findings in their manuscript fully available?

Reviewer #1: (No Response)

Reviewer #2: Yes

5. Is the manuscript presented in an intelligible fashion and written in standard English?

Reviewer #1: Yes

Reviewer #2: Yes

6. Review Comments to the Author

Reviewer #1: All previous comments are taken care off. The revised manuscript can be accepted in its current form.

Reviewer #2: (No Response)

7. PLOS authors have the option to publish the peer review history of their article (what does this mean?). If published, this will include your full peer review and any attached files.

Reviewer #1: No

Reviewer #2: **Yes: **Md. Fahad Jubayer

---

## [Editor Report · Acceptance letter]

26 Feb 2024

PONE-D-23-38533R1 

PLOS ONE

Dear Dr. Shi, 

I'm pleased to inform you that your manuscript has been deemed suitable for publication in PLOS ONE. Congratulations! Your manuscript is now being handed over to our production team.

Kind regards, 

on behalf of

Professor Liang-Jian Deng 

Academic Editor

PLOS ONE